# Automating Dataset Updates Towards Reliable and Timely Evaluation of Large Language Models

**Jiahao Ying[1], Yixin Cao[2]\*, Yushi Bai[3], Qianru Sun[1], Bo Wang [4], Wei Tang[5],**
**Zhaojun Ding[6], Yizhe Yang[4], Xuanjing Huang[2], Shuicheng Yan[7]\***
[1]Singapore Management University, Singapore     [2]Fudan University, China
[3]Tsinghua University, China     [4]Beijing Institute of Technology, China
[5] University of Science and Technology of China, China
[6]School of Computing, University of Georgia, USA     [7]Skywork AI
`{jhying.2022}@phdcs.smu.edu.sg`

## Abstract

Large language models (LLMs) have achieved impressive performance across various natural language benchmarks, prompting a continual need to curate more difficult datasets for larger LLMs, which is costly and time-consuming. In this paper, we propose to automate dataset updating and provide systematical analysis regarding its effectiveness in dealing with benchmark leakage issue, difficulty control, and stability. Thus, once current benchmark has been mastered or leaked, we can update it for timely and reliable evaluation. There are two updating strategies: 1) mimicking strategy to generate similar samples based on original data, preserving stylistic and contextual essence, and 2) extending strategy that further expands existing samples at varying cognitive levels by adapting Bloom's taxonomy of educational objectives. Extensive experiments on updated MMLU and BIG-Bench demonstrate the stability of the proposed strategies and find that the mimicking strategy can effectively alleviate issues of overestimation from benchmark leakage. In cases where the efficient mimicking strategy fails, our extending strategy still shows promising results. Additionally, by controlling the difficulty, we can better discern the models' performance and enable fine-grained analysis — neither too difficult nor too easy an exam can fairly judge students' learning status. To the best of our knowledge, we are the first to automate updating benchmarks for reliable and timely evaluation. Our demo leaderboard can be found at https://yingjiahao14.github.io/Automating-DatasetUpdates/.

## 1 Introduction

Large Language Models (LLMs) are becoming increasingly important in both academia and industry, such as enhancing language translation systems, improving customer service bots, and streamlining data analysis processes across sectors [23, 2, 21, 38, 31, 43]. They have achieved to some extent general intelligence, showing superior performance across various benchmarks including GLUE [35], SQuAD [24], CoQA [25]. As scaling continues, LLMs are gradually mastering more challenging datasets, demanding experts to curate more difficult datasets, which may soon be conquered again by larger and more advanced LLMs. Clearly, such a way of constantly updating dataset is impractical. In this paper, we aim to automate updating benchmarks for LLMs evaluation to minimize human

---

\* Corresponding author.

efforts. This not only helps to timely understand the advantages and disadvantages of model iteration, but also benefits the reliability of evaluation. Once the benchmark leakage issue — testing samples have been seen during pre-training — has been found, we can automatically update current datasets to avoid overestimation. However, this is non-trivial due to the following research questions:

- Will updated benchmarks produce stable results?

- How can the update strategy mitigate benchmark leakage issue?

- Is it possible to automate benchmark updating for better discerning model capabilities?

To this end, we propose two benchmark updating strategies, mimicking and extending a given dataset, and conduct in-depth analysis toward reliable and timely evaluation. i) Our **mimicking** strategy is to leverage LLMs to generate similar ones for each existing sample (marked as seeds), so that we, to the maximum extent, preserve the stylistic and contextual essence of the original data. This is simple and efficient, while it is under exploration if the overestimation caused by data leakage can be mitigated. ii) Inspired by cognitive theory, our **extending** strategy further expands the original data according to varying cognitive levels. Here, we borrow the concepts from Bloom's taxonomy, *a hierarchical classification widely used for educational learning objectives into levels of complexity and specificity.* This not only makes our evaluation more systematic but also leads to a balanced difficulty of the dataset, which can better distinguish and analyze the capabilities of models — neither too difficult nor too easy an exam can fairly judge students' learning status.

In our experiment, we systematically investigate the above two strategies regarding reliability, stability, and their effectiveness to deal with overestimation when benchmark leakage happens. We generate datasets based on two widely used benchmarks (i.e., MMLU [10] and BIG-Bench [5]), and study seven open-source models and four closed-source models. We find that: **1)** Both mimicking and extending strategies show a high level of stability (Section 3.2 & 3.4). **2)** The mimicking strategy proves effective in alleviating overestimation. In most cases, compared to the original leaked dataset, our updated dataset exhibits no significant overestimation issues. (Section 3.3). **3)** In cases where the mimicked dataset still exhibits overestimation, our extending strategy effectively alleviates this issue (Section 3.4). **4)** We can manipulate Bloom's concept of a sample and the popularity of seeds to control the difficulty of the extended dataset. The experimental results also provide a fine-grained analysis of LLMs' cognitive levels. Some models demonstrate considerable variations in their performance across different cognitive levels, yet GPT-4 exhibits a strong performance across all levels (Section 3.5). Our main contributions can be summarized as:

- To the best of our knowledge, we are the first to automate updating benchmarks for timely and stable evaluation.

- We propose to control the difficulty of the sample based on varying cognitive levels, toward fair and fine-grained analysis.

- We have conducted extensive experiments demonstrating the effectiveness of our two strategies in alleviating the issue of overestimation when benchmark leakage occurs.

## 2 Auto-Dataset Update Framework

Figure 1 shows the framework of our auto-dataset update strategies. Given a test sample, mimicking strategy generates one or several similar yet unseen samples, whereas extending strategy generates a set of samples at different cognitive levels. Compared with mimicking strategy, extending strategy is beyond the scope of the given sample, challenging the model's capabilities in more nuanced and complex scenarios. We introduce these two strategies in Section 2.1 and Section 2.2, respectively. Then, we apply them to widely used benchmarks and manually analyze the quality (Section 2.4).

### 2.1 Mimicking Strategy

Given a seed sample and the corresponding task description (optional), we design a prompt for LLMs to generate a new sample that retains the stylistics and knowledge essential. This, to the maximum extent, ensures the quality of generation. Below is an example and more examples are shown in

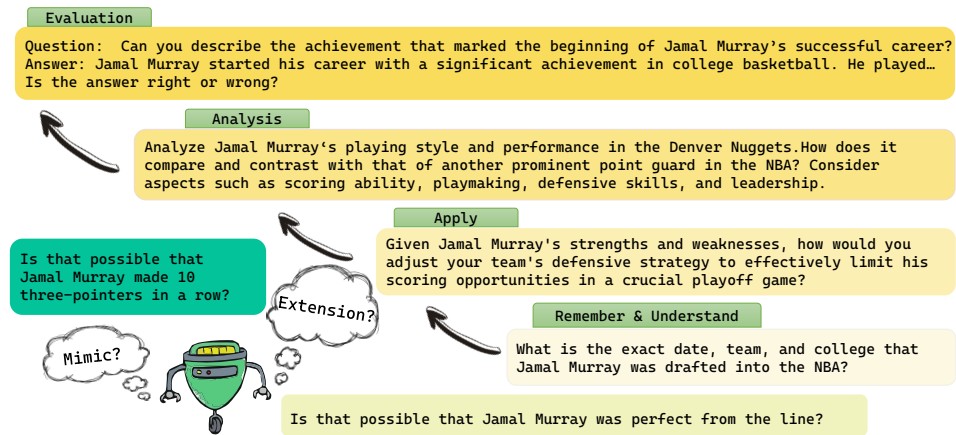

Figure 1: The auto-dataset update framework. For the mimicking strategy, we mimic the `original test case` to get a similar but `new sample` . For the Extending strategy, we extend the original sample to multi-cognition levels ( `Remember & Understand` , `Apply` , `Analysis` , `Evaluation` ) to have a thorough and nuanced assessment.

> *You are a question-writer expert. Please mimic the provided examples to generate \*one\* different but high-quality sample following the task description.*
> *[Task Description]: This task evaluates the model's ability to discern the plausibility of specific athletic actions based on the athlete's known skills and typical behaviors in their sport. For example, a language model should understand that Leo Messi (arguably the best soccer player) is more likely to score goals.*
> *[Seed Sample]: { "input": "Jamal Murray was perfect from the line", "target_scores": { "plausible": 1, "implausible": 0 } }*
> *[New Generated Sample]: {"input": "Jamal Murray made 10 three-pointers in a row", "target_scores": { "plausible": 1, "implausible": 0 } }*

Appendix E. To further improve the quality of generated samples, we heuristically validate them through LLMs themselves or programs to filter out noise, such as answer-incorrect and duplicate samples (details in Appendix A)).

## 2.2 Extending Strategy

Inspired by cognitive theory, an effective learning material should consider different educational objectives. According to Bloom's taxonomy, there are six levels of complexity and specificity: Remember, Understand, Apply, Analyze, Evaluate, and Create. These cognitive levels not only systematically and comprehensively categorize testing data, but also distinguish the difficulty, making it possible to control the difficulty of updated datasets — neither too difficult nor too easy exam can fairly judge students' learning status. Here, we re-organize them into four groups to better fit the purpose of evaluation. As shown in Figure 1, **Remember and Understand Level** asks the model to list, recall basic concepts, interpret, summarize, and exemplify ideas or concepts; **Apply Level** tests the use of learned facts and abstractions in new contexts and particular situations; **Analysis Level** requires the model to break down concepts and examine the relationships among them; **Evaluation Level** asks the model to appraise a situation and criticize opinions or statements. We've combined the Remember and Understand levels for simplicity and excluded the Create level to clearly distinguish between the different cognitive abilities during testing. Following the above idea, we first abstract the original question into a core entity, statement, or piece of knowledge, marked as seeds. Based on that, we then prompt LLMs to generate new questions at different cognitive levels. Below is an example where we extract sports star "Jamal Murray" and prompt LLMs to extend (more examples are shown in Appendix E and the detailed prompt is shown in Appendix D). Clearly, we can adjust the distribution of generation at the four cognitive levels to control the overall difficulty of the updated

dataset. Later, we will provide empirical analysis. It is important to note that our two strategies are efficient and adaptable, making them easily applicable across a wide range of benchmarks.

> *You are a question writer expert, your objective is to write \*\*only one\*\* really complex and difficult question about the given entity.*
> *[Generate Criterion]: 1. The question should be focused on the remember and understand level. This means the question should prompt for recall of facts, terms, and basic concepts, and NOT delve into deeper levels like Applying Analyzing or Understanding. 2. Ensure that you can confidently answer the questions you are proposing, if you can not answer it correctly or have no related knowledge about the entity please return "None".*
> *[Seed Entity]: Jamal Murray*
> *[New Generated Sample]: {"question": What is the exact date, team and college that Murray was drafted into the NBA?, "ref_answer": Answer... }*

### 2.3 Apply to Existing Benchmarks

To conduct auto-dataset update, we select BIG-Bench [5] and MMLU [10] as our seed datasets. From these benchmarks, we chose ten sub-tasks that are particularly representative of the diverse abilities LLMs are expected to demonstrate, ensuring a comprehensive evaluation. These tasks cover various types of questions, including both context-free and context-based, and assess the model's abilities in logical reasoning, common sense reasoning, memorization, mathematical ability, and more. A detailed analysis of the tasks is in Appendix A. Specifically, from BIG-Bench, we include:

**Sports Understanding (Sports):** focuses on the ability to discern between plausible and implausible statements about sports stars. **Periodic Elements (Element):** measures knowledge of chemistry. **CS Algorithms (Algos):** assesses models' understanding of computer science algorithmic. **Physical Intuition (Phys):** tests the understanding of the physical behaviors. **Math Word Problems with Hints (Math):** tests the ability of models to perform mathematical reasoning. From the MMLU, we select: **Abstract Algebra (Algebra):** assesses models' understanding of abstract algebra concepts. **International Law (Law):** evaluates models' ability to understand and follow rules and regulations. **Econometrics (Econ):** tests the understanding of econometric principles and implications of econometric phenomena. **College Medicine (Medicine):** evaluates models' knowledge relevant to medicine. **Computer Security (Security):** involves understanding the principles used to protect computer systems and networks. After selecting datasets, we apply our two strategies to update these datasets and analyze the effectiveness and reliability of the updated samples.

### 2.4 Updated Dataset Analysis

We conduct experiments using GPT [23, 22] series and the Claude [1, 2], series model. However, other LLMs can also be deployed into this framework. For mimicking strategy, we use ChatGPT (gpt-3.5-turbo) and GPT-4-preview to update the selected datasets from BIG-bench and MMLU. Following the update, the filtering process—detailed in Appendix A—may result in a dataset size that is smaller than the original. To achieve a dataset size comparable to the original, we literalize the dataset 2-3 times. (considering the time and the cost we limit the amount of the sample from the dataset "Math" to 1000). The statistical details of the updated datasets are shown in Table 1.

| Task_BIG | #Orig. | #Mimic | Task_MMLU | #Orig. | #Mimic |
|---|---|---|---|---|---|
| Sports | 1000 | 951 | Algebra | 100 | 93 |
| Element | 536 | 548 | Law | 121 | 117 |
| Algos | 160 | 150 | Econ | 114 | 101 |
| Phys | 81 | 81 | Medicine | 172 | 160 |
| Math | 7688 | 1016 | Security | 100 | 100 |

Table 1: The statistical result of the original (Orig.)and mimicked (Mimic) ten subtasks from BIG-Bench and MMLU. For time and cost consideration, we limited the number of the generated samples on Math.

| Metrics | Mimicking | Extending |
|---|---|---|
| Fluency | 94.7 / 95.7 | 98.3 / 100 |
| Coherence | 94.4 / 94.0 | 96.7 / 96.7 |
| Answer Accuracy | 86.7 / 82.8 | 92.7 / 82.0 |
| Category Accuracy | - | 98.3 / 100 |

Table 2: The overview of Human Evaluation. The Scores are shown as full score rate (%), with numbers after the slash indicating agreement rates (%) among the five evaluators.

For the extending strategy, we chose the dataset Sports, Algorithms (Algos), Algebra, and Physics (Phys), because the mimicking strategy performs poorly for benchmark leakage mitiga-

tion(Section 3.3). For the Sports task, we extract the names of sports stars as seeds, recognizing that this task assesses the models' knowledge of these individuals. For the Algebra task, we extract key algebraic concepts as seeds. For Phys, we use GPT-4 [23] to summarize the basic physical laws behind the original samples as seeds. Since the Algo task only encompasses a limited range of algorithmic topics, we employ GPT-4 to generate a list of 40 algorithm names. These names are then utilized as seeds for question generation. Based on these seeds, we utilize GPT-4 and Claude-3 [2] to generate new samples across multiple cognitive levels using extending prompts (details in Appendix D). We manually maintain an equal distribution of the questions across each cognitive level. The static result is shown in Table 11.

**Human Evaluation.** To validate the reliability of our two strategies, we conduct a human evaluation involving five senior computational linguistics researchers, who have been trained in advance. For mimicked samples, evaluators review 120 randomly chosen samples, each including the question's category, the question, and the answer. They assess each question and answer paired based on three criteria: Fluency (the grammatical correctness and smoothness of the question), Coherence and Clarity (the logical clarity and explicit articulation of the question), and Accuracy of the Answer (the detailed evaluation guideline is shown in Appendix B.1). In evaluating the extended samples, evaluators examine 60 randomly selected samples, which include the question, its cognitive level, and the reference answer. The assessment criteria are similar to those for the mimicked samples, with an additional focus on Category Accuracy (the detailed evaluation guideline is shown in Appendix B.2) to ensure that the question's cognitive level is appropriately identified. The human evaluation results, summarized in Table 2, demonstrate high effectiveness and reliability of both strategies.

## 2.5  Evaluation Metric

For mimicked samples, we adhere to the evaluation metrics used in the original tasks (details in Appendix C.1.5). For extended samples, where questions are free-form, we adopt the "LLM judgment" methodology, following [47, 3]. We give the question, reference answer (from updated sample) and the candidate's answer to LLMs to evaluate the answer across three dimensions: 1. **Accuracy:** evaluates the correctness of the answer, 2. **Coherence:** assesses the logical flow, and 3. **Factuality:** assessing the presence of factual errors (the evaluation prompt is shown in Appendix D). We use the full-mark rate over the three dimensions as the metric and manually evaluate the "LLM judgment" result. The consistency of the model's results with human judgment achieved 90.8% (detailed evaluation guideline is in Appendix B.2 and detailed human evaluation result is shown in Table 8).

# 3  Experiment

## 3.1  Baseline

For baselines, we choose seven open source models: Llama-2-7b-chat, Llama-2-13b-chat [33], Llama-3-8b-Instruction [20], Mistral-7B-Instruct-v0.2, Mixtral-8x7B-Instruct-v0.1 [13], Yi-6b-chat, Yi-34b-chat [16], and four closed-source models: GTP-4, ChatGPT, Claude2 [1], Gemini-pro [9].

## 3.2  Stability of Mimicked Datasets

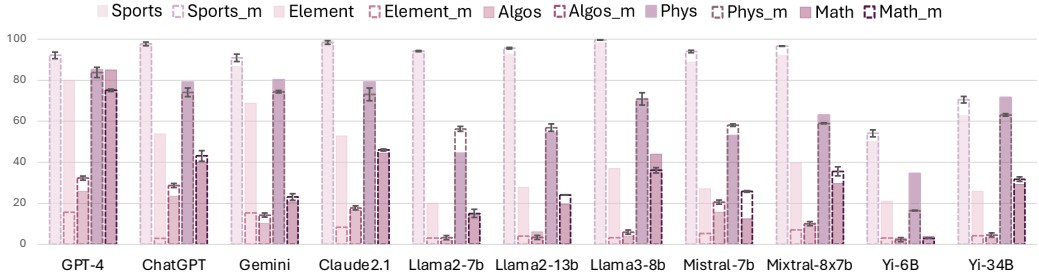

Figure 2: Performance (%) of the 11 involved models (zero-shot) on the original and mimicked (footnote $m$) Big-Bench. The generation of the mimicked dataset is conducted four times, the figure displays the average performance and standard deviation. Detailed results are shown in Table 12.

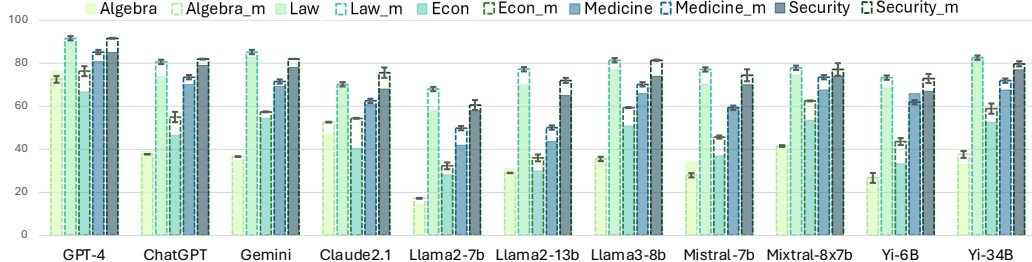

Figure 3: Performance (%) of the 11 involved models (zero-shot) on the original and mimicked MMLU datasets. More detailed results are presented in Table 13.

Our proposed strategies are able to update datasets with minimum human effort whenever needed. That is, if we apply our strategy to the same dataset several times, we will obtain multiple different datasets. Clearly, a major concern is whether they can produce consistent evaluation results. To answer the above stability question, we choose the mimicking strategy and iterate the update process four times. Similar experiments under the extending strategy will be discussed in Section 3.4. The statistical details can be found in Table 9. Based on four mimicked datasets, we calculate the average performance and the standard deviation of 11 baseline models under zero-shot in context learning (ICL). As shown in Figure 2 and Figure 3 (more detail is shown in Table 12 and Table 13) It is apparent that: **1)** Compared with the performance of baselines on original datasets, their performance on our curated datasets is similar — the difference of the two scores is 5% on average except Element (we will discuss it later) , 18% max and 0% min. This is mainly because the mimicked dataset has a similar style and difficulty; **2)** Among four different mimicked datasets, the standard deviation is limited, ranging from 0% to 3%. This demonstrates the stability of our mimicking dataset update strategy; **3)** The performance on the "Periodic Elements" in the mimicked dataset is markedly lower than in the original. This discrepancy can be attributed to the inclusion of the periodic table in the task description to assist the model in generating new questions. Unlike the original task, which exhaustively covered all possible one-hop questions like *"What element contains one more proton than hydrogen?"*, the model, with the aid of the periodic table, generates more complex queries such as *"What element contains two more protons than hydrogen?"*, which may need a reasoning step to answer compared to the original sample. This also demonstrates the possibility of introducing external knowledge for updates, which will be future work.

### 3.3 Can mimicked datasets alleviate overestimation when benchmark leakage occurs?

| Model | Training | LoRA | $Sports_o$ | $Sports_m$ | $Element_o$ | $Element_m$ | $Algos_o$ | $Algos_m$ | $Phys_o$ | $Phys_m$ | $Math_o$ | $Math_m$ |
|---|---|---|---|---|---|---|---|---|---|---|---|---|
| Llama2-7b | None | - | 94.3 | 94.2 | 19.9 | 3.1 | 2.0 | 2.6 | 44.4 | 57.5 | 14.6 | 15.1 |
| Llama2-7b | + leakage | ✓ | 99.7 | 87.4 | 57.1 | 0.9 | 42.2 | 34.0 | 74.9 | 51.3 | 43.6 | 18.0 |
| Llama2-7b | + w rationale | ✓ | 92.4 | 92.7 | 39.9 | 1.3 | 37.2 | 36.0 | 67.9 | 57.5 | 25.8 | 19.8 |
| Llama2-7b | + leakage | ✗ | 99.8 | 85.8 | 42.9 | 0.0 | 32.5 | 25.3 | 70.4 | 52.5 | 42.0 | 19.2 |
| Llama2-7b | + w rationale | ✗ | 99.7 | 88.7 | 47.8 | 0.2 | 40.6 | 34.0 | 70.4 | 55.6 | 32.4 | 20.8 |
| Llama2-13b | None | - | 92.7 | 96.1 | 27.8 | 4.0 | 6.1 | 3.4 | 54.3 | 58.5 | 19.6 | 24.6 |
| Llama2-13b | + leakage | ✓ | 99.8 | 89.2 | 65.7 | 1.1 | 54.4 | 42.7 | 83.9 | 61.3 | 43.6 | 23.1 |
| Llama2-13b | + w rationale | ✓ | 96.5 | 91.7 | 49.3 | 1.8 | 45.6 | 43.3 | 74.0 | 55.0 | 32.0 | 28.6 |
| Llama2-13b | +leakage | ✗ | 99.7 | 88.5 | 40.7 | 0.0 | 36.3 | 28.0 | 71.6 | 45.0 | 42.3 | 26.4 |
| Llama2-13b | + w rationale | ✗ | 94.3 | 88.9 | 43.6 | 0.9 | 37.3 | 39.4 | 76.6 | 62.5 | 36.4 | 28.2 |
| Llama-8b | None | - | 98.2 | 99.7 | 36.9 | 3.3 | 4.1 | 6.0 | 70.4 | 67.5 | 43.9 | 34.8 |
| Llama3-8b | + leakage | ✓ | 98.7 | 92.5 | 66.2 | 2.6 | 36.6 | 39.3 | 77.8 | 67.5 | 57.1 | 34.4 |
| Llama3-8b | + w rationale | ✓ | 98.1 | 87.7 | 68.2 | 7.5 | 39.4 | 44.0 | 86.9 | 71.3 | 51.5 | 37.1 |
| Llama3-8b | + w rationale | ✗ | 93.2 | 85.6 | 60.8 | 12.4 | 36.6 | 39.3 | 79.0 | 66.3 | 44.5 | 29.9 |
| Mistral-7b | None | - | 88.8 | 94.0 | 27.1 | 5.3 | 15.7 | 20.0 | 53.1 | 57.5 | 12.5 | 25.8 |
| Mistral-7b | + leakage | ✓ | 99.8 | 87.7 | 36.6 | 1.6 | 38.1 | 30.7 | 66.7 | 58.8 | 49.9 | 24.1 |
| Mistral-7b | + w rationale | ✓ | 95.0 | 90.4 | 54.8 | 1.6 | 46.6 | 40.6 | 81.0 | 58.8 | 34.4 | 21.5 |
| Mistral-7b | + w rationale | ✗ | 98.3 | 88.2 | 61.0 | 0.0 | 45.9 | 38.0 | 88.9 | 56.8 | 48.0 | 27.5 |

Table 3: Finetune performance (%) (zero-shot) of Llama-2-7b-chat, Llama-2-13b-chat, Llama-3-8b-Instruction and Mistral-7B-Instruct on the original and mimicked BIG-bench examples , *leakage* denote using test prompt and the test set during training. *w rationale* denote using test set with rationale (Sec 3.3). For our fine-tuning process, we employed: full parameter and LoRA-only. Cells are blue if finetuning boosts the performance less than 5% else are in red. Specifically, less than 0% , more than 0 % but less than 5% , more than 5 % but less than 10% , more than 10 % .

| Model | Training | LoRA | $\text{Algebra}_o$ | $\text{Algebra}_m$ | $\text{Law}_o$ | $\text{Law}_m$ | $\text{Econ}_o$ | $\text{Econ}_m$ | $\text{Medicine}_o$ | $\text{Medicine}_m$ | $\text{Security}_o$ | $\text{Security}_m$ |
|---|---|---|---|---|---|---|---|---|---|---|---|---|
| Llama2-7b | None | - | 14.0 | 17.2 | 57.8 | 70.0 | 28.1 | 32.7 | 41.9 | 49.8 | 58.0 | 60.0 |
| Llama2-7b | + leakage | ✓ | 52.0 | 31.2 | 95.9 | 70.8 | 65.8 | 34.6 | 79.8 | 50.4 | 86.0 | 66.0 |
| Llama2-7b | + w rationale | ✓ | 33.0 | 30.1 | 74.4 | 72.7 | 38.6 | 31.7 | 53.8 | 50.8 | 67.0 | 60.0 |
| Llama2-7b | + leakage | ✗ | 49.0 | 23.7 | 93.4 | 70.9 | 65.8 | 33.6 | 79.2 | 51.1 | 84.0 | 60.0 |
| Llama2-7b | + w rationale | ✗ | 42.0 | 31.2 | 81.8 | 73.5 | 46.5 | 36.6 | 62.4 | 55.0 | 76.0 | 63.0 |
| Llama2-13b | None | - | 31.0 | 29.0 | 70.0 | 77.8 | 30.0 | 36.7 | 43.6 | 50.1 | 65.0 | 72.0 |
| Llama2-13b | +leakage | ✓ | 51.0 | 30.1 | 96.7 | 79.5 | 67.5 | 42.5 | 83.2 | 54.9 | 92.0 | 76.0 |
| Llama2-13b | + w rationale | ✓ | 34.0 | 32.6 | 86.0 | 80.5 | 39.5 | 40.2 | 58.4 | 55.5 | 75.0 | 74.0 |
| Llama2-13b | +leakage | ✗ | 48.0 | 23.7 | 92.6 | 70.9 | 60.5 | 36.6 | 82.1 | 50.4 | 83.0 | 71.0 |
| Llama2-13b | + w rationale | ✗ | 38.0 | 35.4 | 86.7 | 80.1 | 50.0 | 40.5 | 65.3 | 55.6 | 79.0 | 74.0 |
| Llama3-8b | None | - | 34.0 | 36.5 | 76.9 | 82.0 | 50.9 | 59.4 | 65.9 | 70.3 | 74.0 | 81.0 |
| Llama3-8b | +leakage | ✓ | 49.0 | 29.0 | 92.6 | 83.7 | 72.8 | 61.3 | 87.2 | 75.0 | 89.0 | 80.0 |
| Llama3-8b | + w rationale | ✓ | 48.0 | 38.7 | 88.4 | 83.7 | 65.8 | 63.4 | 74.6 | 71.8 | 88.0 | 84.0 |
| Llama3-8b | + w rationale | ✗ | 54.0 | 39.7 | 90.1 | 87.2 | 74.6 | 61.3 | 76.3 | 75.6 | 80.0 | 83.0 |
| Mistral-7b | None | - | 34.0 | 26.8 | 70.2 | 77.1 | 36.9 | 45.7 | 59.5 | 59.3 | 70.0 | 74.0 |
| Mistral-7b | + leakage | ✓ | 63.0 | 32.6 | 96.7 | 75.2 | 70.2 | 42.6 | 90.2 | 56.3 | 90.0 | 75.0 |
| Mistral-7b | + w rationale | ✓ | 39.0 | 30.1 | 90.0 | 79.3 | 54.4 | 46.5 | 75.7 | 61.8 | 78.0 | 76.0 |
| Mistral-7b | + w rationale | ✗ | 50.0 | 26.9 | 97.5 | 80.1 | 68.4 | 50.8 | 79.8 | 62.1 | 86.0 | 76.0 |

Table 4: Finetune performance (%) of Llama-2-7b-chat, Llama-2-13b-chat, Llama-3-8b-Instruction and Mistral-7B-Instruct on the mimicked MMLU examples. Following the setting in Table 3.

One of our main motivations is to automate dataset updates toward reliable evaluation. In this section, we validate how our approach mitigates the overestimation caused by benchmark leakage. We choose four baseline models, Llama-2-7b-chat, Llama-2-13b-chat [33], Llama-3-8b-Instruction [20] and Mistral-7B-Instruct [12], and follow the training configuration in previous work [48, 33, 42] for fair evaluation (detailed configuration is in the Appendix C). For each model, there are three training settings: **1)** *None*, which denotes the original model; **2)** *leakage*, referring to the finetuned model using the above leakage simulation; **3)** *w rationale*, which includes rationales for each leakage sample during finetuning. Thus, these rationale finetuning will also improve the models' reasoning ability over the testing samples. In specific, we follow [37] to generate rationales using GPT-4 [23] for the correct choice. The detailed prompt is shown in the Appendix D.3. As shown in Table 3 and Table 4, we find that: **1)** on the original dataset, the performance of all leakage simulations increases dramatically, no matter finetune via LoRA [11] or not. This is consistent with previous works [48], showing the serious overestimation issue; **2)** on our mimicked datasets, the performance of all leakage simulations generally decreases or remains similar, except task Algos, and Algebra (discussed with the extending strategy Sec 3.4); **3)** in most cases, the performance of *leakage* setting is better than that of *w rationale* on original datasets, while the results are opposite in mimicked datasets. We attribute this to the improvement of the generalization ability through training with rationales, which will mitigate the overfitting performance of *leakage/w rationale*, aligning with [17]. **4)** Llama-2-13b improves from 3.4% to 43.3% on the task Algos after finetuning. We hypothesize this improvement is attributable to model learning the distribution of labels, considering the labels range uniformly from 1 to 10. This indicates a limitation of the mimicking strategy: the possibility of benchmark leakage leading to models learning from the label distribution, which leads to an overestimated performance.

## 3.4 Extending Strategy: Stability & Dealing with benchmark Leakage

As shown in Section 3.3, the mimicking strategy does not work well on two datasets: Algos and Algebra. The benchmark leakage issue still leads to a severe overestimation of the model, which we will further explore using the extending strategy in this section. Following the experiment settings of the mimicking strategy, we leverage the same baseline models with leakage simulations, as well as four iterations of dataset updates for stability verification (updating process defined in Section 2.2 and Section 2.4 and statistical details can be found in Table 10). As shown in Table 5, the standard deviation among four runs is quite low, demonstrating the stability of our extending strategy. This is not surprising as extending strategy can be regarded as an extension of the mimicking strategy with varying cognitive levels for controllable difficulty (Section 3.5). To investigate the effectiveness, we employ baseline models in the mimicking strategy that are fine-tuned on test samples from the original datasets. Table 5 shows that all performances of leakage simulations greatly decrease compared with original models due to the overfitting issue. Moreover, the decline of *leakage* is more than that of *w rationale* because it is alleviated through the improved generalization ability via learning with rationales. However, note that even with rationales, the model still exhibits signs of overfitting, which may improve its performance on the original test set. We also calculate the similarity between the extended data and data from public sources following previous work [15, 7], and our result shows that extending strategy won't have a benchmark leakage issue (more detail is shown in Appendx C.2.2).

| Model | Training | LoRA | Algebra | Algos |
|-------|----------|------|---------|-------|
| Llama2-7b | None | - | **4.2** ±0.6 | **10.2** ±0.5 |
| Llama2-7b | + leakage | ✓ | 1.6 ±0.6 | 2.2 ±0.6 |
| Llama2-7b | + w rationale | ✓ | 1.2 ±0.0 | 5.3 ±0.5 |
| Llama2-7b | + leakage | ✗ | 1.8 ±0.6 | 1.1 ±0.0 |
| Llama2-7b | + w rationale | ✗ | 2.1 ±0.5 | 8.9 ±0.8 |
| Llama2-13b | None | - | **8.5** ±0.4 | **11.9** ±0.8 |
| Llama2-13b | + leakage | ✓ | 6.6 ±0.5 | 5.1 ±0.1 |
| Llama2-13b | + w rationale | ✓ | 6.4 ±0.9 | 11.2 ±0.8 |
| Llama2-13b | + leakage | ✗ | 1.5 ±0.4 | 0.5 ±0.4 |
| Llama2-13b | + w rationale | ✗ | 5.7 ±0.5 | 7.7 ±1.2 |
| Llama3-8b | None | - | **34.6** ±1.6 | **46.7** ±2.1 |
| Llama3-8b | + leakage | ✓ | 12.1 ±0.5 | 21.3 ±2.2 |
| Llama3-8b | + w rationale | ✓ | 17.8 ±1.3 | 17.5 ±1.8 |
| Llama3-8b | + w rationale | ✗ | 15.3 ±1.6 | 18.7 ±1.2 |
| Mistral-7b | None | - | **21.8** ±1.7 | **36.8** ±0.0 |
| Mistral-7b | + leakage | ✓ | 0.9 ±0.5 | 5.3 ±0.5 |
| Mistral-7b | + w rationale | ✓ | 4.4 ±0.6 | 9.1 ±0.5 |
| Mistral-7b | + w rationale | ✗ | 2.3 ±0.2 | 4.4 ±0.8 |

Table 5: Average full-mark (%) of the fine-tuned model on the extended dataset over the four iterations. Blue cells indicate reduced performance after fine-tuning.

### 3.5 Is it possible to control the sample difficulty using the Extending Strategy?

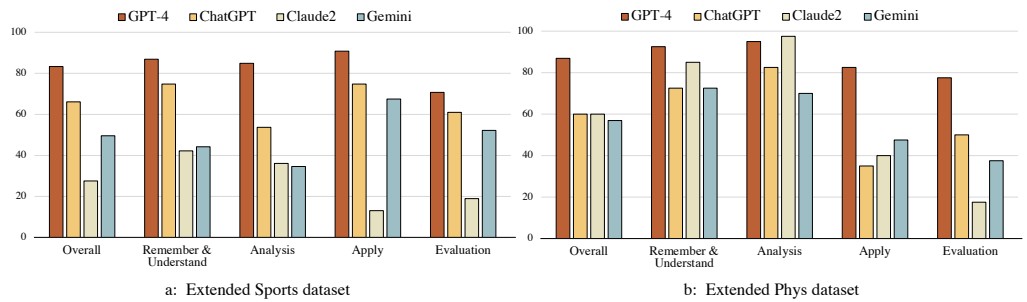

a: Extended Sports dataset      b: Extended Phys dataset

Figure 4: Full-mark rate (%) of GPT-4, ChatGPT, Claude, and Gemini on the extended Sports (a), Phys (b) dataset on different cognitive levels. Overall is the average score across the four levels.

Except for overestimation leading by benchmark leakage in Section 3.4, there is another issue on the task Sports and Phys (Table 3 and Table 4) — the absolute scores are notably high, suggesting that the questions are relatively simple and thus insufficient to differentiate the models (i.e., 3.82% and 4.30% difference on average for models GPT-4, ChatGPT, Gemini, and Claude). In this section, we explore the feasibility of modulating the difficulty of the extended data samples to better discern and differentiate model capabilities. In our extension work, we start by abstracting the original question into core entities, statements, or adding additional knowledge, then use Bloom's taxonomy for further extension. This naturally allows for the adjustment of question difficulty in two ways, as indicated by prior findings: 1) work [14] suggests that cognitive demand increases with higher cognitive processes. Accordingly, by adjusting to more abstract cognitive levels, we expect to produce more challenging samples; 2) work [18] indicates that as the popularity of the subject entity increases, the difficulty of the question decreases. Thus, the popularity of the input seed could be strategically manipulated to adjust the difficulty of the generated questions.

To validate the efficacy of our framework in controlling difficulty, we use the extracted sports star name and the summarized common physical law as the seed to update the dataset using the extending strategy (Sec 2.4). We use models GPT-4, ChatGPT, Gemini, and Claude as the baseline for their indistinguishable performance. As shown in Figure 4, **1)** on the extended Sports and Phys datasets, the four models' performance show larger variance (23.76%, 14.04% respectively), compared with original datasets; **2)** overall, samples test "Apply" and "Evaluation" cognitive levels are found to be more challenging. This finding indicates that manipulating the cognitive level in the extending strategy is an effective way of controlling the difficulty of the generated questions in our framework. It also highlights the importance of focusing on models' capabilities at different cognitive levels.

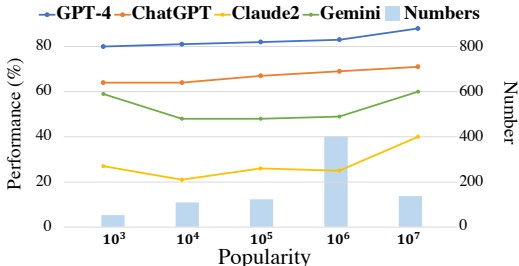

Figure 5: Full-mark rate (%) of GPT-4, ChatGPT, Claude and Gemini on the extended "sport understanding" datasets on different popularity levels. We depict the number of questions on different popularity levels. Despite an uneven distribution for popularity, we still can find that as the popularity of the subject entity increases, the difficulty of the question increases

Some models exhibit significant disparities in performance at various cognitive levels, while GPT-4 stands out with its superior and consistent performance.

Besides assessing how cognitive levels influence performance, we explore how the popularity of the seeds used in dataset updates affects model performance. Following previous work [18], where using Wikipedia page views as the popularity, we gather the total page views for each seed throughout year 2023 and use it as the popularity. As in Figure 5, despite an uneven distribution of seed popularity in the extended dataset, a clear trend emerged: there was a noticeable increase in model performance when using more popular seeds to generate the question. This suggests that the popularity of the seed input can also be strategically manipulated to control the difficulty in our framework.

### 3.6 Adaptable to Models Beyond the GPT Backbone

Relying solely on GPT-4 may introduce biases, such as self-preference, particularly when employing the "LLM-as-an-Examiner" methodology [3]. In this section we expand the experimentation by using Claude-3-Opus [2] to generate new samples, demonstrating the framework's adaptability to other backbones. Thus by using multi-source data, we can mitigate the bias associated with exclusively using GPT-4. Using Claude-3-Opus, we extend the algebra task, where the mimicking strategy is less effective, following the setting in Section 3.3. Due to time and cost constraints, we conduct the generation process twice. As shown in Table 10, we observe a consistent conclusion with Table 5: the performance of models on the leaked original dataset declines when evaluated on our extended dataset. This indicates the adaptability of our framework to utilize different language models as backbones.

| Model | Training | LoRA | Algebra |
|-------|----------|------|---------|
| Llama2-7b | None | - | **26.3** ±1.6 |
| Llama2-7b | + w rationale | ✓ | 8.9 ±0.5 |
| Llama2-7b | + w rationale | ✗ | 22.1 ±0.5 |
| Llama2-13b | None | - | **33.9** ±1.9 |
| Llama2-13b | + w rationale | ✓ | 26.0 ±1.0 |
| Llama2-13b | + w rationale | ✗ | 9.7 ±1.6 |
| Llama3-8b | None | - | **59.0** ±1.5 |
| Llama3-8b | + w rationale | ✓ | 39.6 ±0.4 |
| Llama3-8b | + w rationale | ✗ | 41.1 ±1.0 |
| Mistral-7b | None | - | **45.5** ±0.4 |
| Mistral-7b | + w rationale | ✓ | 17.9 ±0.4 |
| Mistral-7b | + w rationale | ✗ | 6.8 ±2.2 |

Table 6: Average performance (%) of the fine-tuned models on the extended datasets generated by Claude-3-Opus over the two iterations. Blue cells indicate reduced performance after fine-tuning.

## 4 Related Work

**Benchmark for Model Evaluation:** There are a lot open-source benchmarks: MMLU [10], tests a wide range of knowledge and reasoning abilities. HellaSwag [46], challenges models with complex commonsense reasoning. BIG-bench [5], tests models on both traditional NLP tasks and novel problems. ARC [6], focusing on science questions that require deep reasoning. KoLA [44], also uses the bloom taxonomy to construct the dataset. Given that these public benchmarks are open-source and static, they are susceptible to benchmark leakage. Recently, there have been efforts to construct benchmarks using newly emerged corpora[32, 41]. EvoWiki [32] categorizes Wikidata and Wikidata into three levels according to the cut-off date of model development and further exam models' performance without leakage. However, whether these context-based question generation methods can generate more challenging questions to better differentiate between models has yet to be proven.

**Data Leakage:** Data Leakage could be a crucial problem, study [48, 28] shows that a fine-tuned model can achieve perfect performance on benchmarks. To combat this, researchers have developed various detection methods [7, 29, 19, 36, 8, 34]. Study [7] proposed detecting exact matches between test examples and pretraining data. Work [15], proposed a contamination report for the open-sources model. The work [39] design two indicators using the perplexity to indicate the potential data leakage. However, these detection methods have limitations for they can not be applied to those closed-source models. In the same period of time [45] use the following-up questions to alleviate data leakage.

## 5 Conclusion

This paper presents two strategies for automating dataset updates toward reliable and timely LLM evaluation. The mimicking strategy generates new, similar samples based on existing ones, while the extending strategy further expands the generated sample using cogitation levels. Extensive experiments using eleven LLMs on updated samples from MMLU and BIG-Bench datasets indicate the stability of our strategies and effectiveness toward addressing benchmark leakage. In the future, we are interested in introducing external or domain-specific knowledge for dataset updates.

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

# A Data analysis

Tabel 7 shows the detailed information of the selected 10 tasks of BIG-bench and MMLU.

| Task | :≡ Label in BIG/ MMLU | Cogitative Level | Validation Method |
|------|----------------------|------------------|-------------------|
| Sports Understanding | context-free question answer
domain specific
common sense
multiple choice  json | Evaluation | - Model-self check |
| Periodic elements | context-free question answer
memorization
domain specific
free response  json | Remember | - Program check |
| CS Algorithms | context question answer
domain specific
free response  json | Apply | - Program check |
| Physical Intuition | context-free question answer
domain specific
multiple choice  json | Apply | - Model-self check |
| Math Word Problems with Hints | context-free question answer
logical reasoning
multiple choice  json | Apply | - Program check |
| Abstract Algebra | context-free question answer
domain specific
multiple choice  json | Apply
Evaluation | - Model-self check |
| Internation Law | context-free question answer
domain specific
multiple choice  json | Remember
Evaluation | - Model-self check |
| Econometrics | context-free question answer
domain specific
multiple choice  json | Remember
Apply
Evaluation | - Model-self check |
| College Medicine | context-free question answer
domain specific
multiple choice  json | Remember
Apply | - Model-self check |
| Computer Security | context-free question answer
domain specific
multiple choice  json | Remember
Apply | - Model-self check |

Table 7: This table elaborates on the detailed task information, question format, and corresponding cognitive levels of the selected tasks from BIG-bench and MMLU (for MMLU we manually labeled each task). Additionally, it outlines the validation method employed for the mimicked tasks. For the "Model-self" check, we deploy the model itself to identify and filter out instances with incorrect answers. For the "Program check" we write programs to filter out instances. The human evaluation results for the mimicked and extended samples are shown in Section 2.4

# B Human Evaluation

## B.1 Mimiced Benchmark Evaluation Guideline

To validate the quality of the mimicked data sample, we randomly selected 120 data samples from the mimicked BIG-bench and MMLU. and Then we conducted a human evaluation involving five senior computing language researchers. Evaluators are provided with samples that have: 1) the question category, 2) the question, and 3) the answer. The evaluation guideline is provided below:

This document is a guide for evaluating test questions generated by Mimic, including the category of the question, the question itself, and the answer. You are tasked with evaluating, Please score based on the following criteria:

**For the question**, using the following metric for scoring:

**Fluency**: Assess whether the language of the question is fluent, without grammatical or spelling errors. The scoring range is 1-3:
- 1 point: The text contains multiple grammatical and/or spelling errors, significantly impacting the readability and understanding.
- 2 points: The text contains a few grammatical or spelling errors, slightly affecting readability, but the overall meaning of the text is understandable.
- 3 points: The text is grammatically and orthographically correct, expressing fluently and naturally, easy to understand.

- **Coherence and Clarity**: Assess whether the question is logically clear and articulated explicitly. The scoring range is 1-3:
- 1 point: The question or answer lacks logical structure, is expressed in a disorganized manner, making it difficult for readers to understand.
- 2 points: The question or answer has a basic logical structure, with a relatively clear theme or argument, but the expression may not be direct enough or some parts may be slightly vague, affecting overall clarity.
- 3 points: The question or answer has a clear structure, is logically coherent, expressed directly and clearly, easy to understand, and effectively conveys the theme or argument.

- **Highly misleading**: The context strongly influences you to choose the answer.

**For the answer, evaluate the Accuracy:**

**Accuracy**: Assess the options and answer. The scoring range is 1-3
- 1 point: The information in the answer does not match the question's requirements, contains serious errors or misleading information, does not answer the question at all, or the options do not have a correct answer.
- 2 points: The answer attempts to address the question but contains partial errors or misses important information.
- 3 points: The answer is completely accurate, appropriately addressing the question.

## B.2 Extend Benchmark Evaluation Guideline

To validate the quality of the extended data sample and the evaluation result, we randomly selected 60 data samples from the extended Sports, Algebra, Algos, and Phys tasks. Then we conduct a human evaluation involving five senior computing language researchers. Evaluators are provided include: 1) question, 2) cognitive level, 3) reference answer, 4) candidate response, and 5) evaluation result. The evaluation guideline is provided below:

- This document is a guide for evaluating test questions generated by Extend method, along with the evaluation result. Each line includes the question "level", the corresponding question, the reference answer, the candidate response, and the evaluation result. Evaluation should be conducted according to the following requirements:

**For the question**, using the following metric for scoring:

**1. Fluency**: Assess whether the language of the question is fluent, without grammatical or spelling errors. The scoring range is 1-3:
- 1 point: The text contains multiple grammatical and/or spelling errors, significantly impacting the readability and understanding.
- 2 points: The text contains a few grammatical or spelling errors, slightly affecting readability, but the overall meaning of the text is understandable.
- 3 points: The text is grammatically and orthographically correct, expressing fluently and naturally, easy to understand.

- **2. Coherence and Clarity:** Assess whether the question is logically clear and articulated explicitly. The scoring range is 1-3:
  - 1 point: The question or answer lacks logical structure, is expressed in a disorganized manner, making it difficult for readers to understand.
  - 2 points: The question or answer has a basic logical structure, with a relatively clear theme or argument, but the expression may not be direct enough or some parts may be slightly vague, affecting overall clarity.
  - 3 points: The question or answer has a clear structure, is logically coherent, expressed directly and clearly, easy to understand, and effectively conveys the theme or argument.
- **3. Category Accuracy:** Based on cognitive theory, questions are categorized as follows:
  - Remember and Understand Level asks the model to list, recall basic concepts, interpret, summarize, and exemplify ideas or concepts.
  - Apply Level tests the model to use learned facts and abstractions in new contexts and particular situations.
  - Analysis Level requires the model to break down concepts and examine the relationships among them.
  - Evaluation Level asks the model to use knowledge and skills to appraise a situation and criticize opinions or statements.
  Score the category accuracy considering if the question category matches the content, ranging from 1-3:
  - 1 point: The category does not match the question content at all. The selected category has no relevance to the question's theme or the type of information required.
  - 2 points: The category partially matches the question content. The selected category is somewhat related to the question but is not the best or most accurate category.
  - 3 points: The category perfectly matches the question content. The selected category precisely reflects the content of the question.

For the provided candidate reponse, you are tasked with evaluating the candidate answers according to three distinct metrics: Accuracy, Coherence, and Factuality. Each of these metrics is critical for assessing the quality of the answers. Below is a detailed scoring guide for each:

1. Accuracy: - 1-3 points: Score from 1 to 3 based on whether the answer is completely inaccurate, partially accurate but contains some errors or misses important information, or is completely accurate, appropriately, and comprehensively answering the question without any errors or omissions.

2. Coherence: - 1-3 points: Score based on whether the answer lacks logical structure and is expressed in a disorganized manner, has basic logical structure and clarity with some unclear or vague parts, or has a clear structure, is logically coherent, expressed clearly, and effectively conveys the theme or argument.

3. Factuality: - 1-3 points: Score based on whether the answer contains multiple factual errors, generally conforms to facts but contains minor errors or inaccuracies, or is entirely based on facts with all provided information being accurate.

**Compare your scores for accuracy, coherence, and factuality with the evaluation results** for

  **Consistency**, ranging from 1-3:
  - 1 point: The model's evaluation results significantly diverge from the actual quality of the question and answer, showing a high degree of mismatch.
  - 2 points: The model's evaluation results reflect the quality of the question and answer to some extent but have some inconsistencies.
  - 3 points: The model's evaluation results are highly consistent with the quality of the question and answer, accurately reflecting its performance.

### B.3 Huaman Evaluation result

| Setting | Fluency | Coherence | Answer Accuracy | Category Accuracy | Evaluation Consistency |
|---|---|---|---|---|---|
| **Mimic** | 94.7 / 95.7 | 94.4 / 94.0 | 86.7 / 82.8 | - | - |
| **Extend** | 98.3 / 100 | 96.7 / 96.7 | 92.7 / 82.0 | 98.3 /100 | 90.8 / 86.7 |

Table 8: This table presents the overview of Human Evaluation. Results Scores are shown as full score rate (%), with numbers after the slash indicating agreement rates (%) among the five evaluators.

Shown in Table 8, human evaluation result indicates over 85% full score rate for each metric. This demonstrates our framework's capability to support reliable automatic question generation.

## C  Experiment detail

### C.1  Model Setting

#### C.1.1  Question Generation

For generating questions, we configure both ChatGPT and GPT-4 with a temperature setting of 0 and a maximum token length of 512 to ensure precise and deterministic output.

#### C.1.2  Answer Generation

Answer generation across the 10 models involved is conducted in a zero-shot setting, with all models set to a temperature of 0 and a maximum token length of 1024. The specific prompt used for generating answers is detailed in Appendix D.

#### C.1.3  LoRA Fine-tuning

For LoRA fine-tuning, we explore a range of configurations, adjusting the rank (from 32 to 1024 for the 7b and 13b models), learning rate (from $10^{-4}$ to $10^{-6}$), number of epochs (from 1 to 3), and batch size (from 2 to 32). The optimal configuration for each model is determined through a grid search strategy. The experiment is conducted on 8×Nvidia A100 GPUs. Our most resource-intensive experiment takes 20 A100 GPU hours.

#### C.1.4  Full Parameter Fine-tuning

Similar to LoRA fine-tuning, full parameter fine-tuning involves adjusting the learning rate (from $10^{-4}$ to $10^{-6}$), number of epochs (from 1 to 3), and batch size (from 2 to 32), with the best performance for each model selected via a grid search strategy. The experiment is conducted on 8×Nvidia A100 GPUs. Our most resource-intensive experiment takes 30 A100 GPU hours.

#### C.1.5  Detailed Metric for mimicking

For the mimic setting, we adhere to the evaluation metrics used in the original tasks considering the similar sample format. Specifically, for multi-choice questions, we utilize the Exact Match (EM) metric to gauge accuracy, for free response questions we Exact Match (EM) metric to gauge accuracy.

#### C.1.6  Detailed Metric for extending

In the extending setting, where questions are free-form, we adopt the "LLM judgment" methodology [47, 3]. We evaluate model performance across three dimensions: 1. **Accuracy:** evaluates the correctness of the answer, 2. **Coherence:** assesses the logical flow and clarity of the response, and 3. **Factuality:** assessing the presence of factual errors in the response (detailed prompt are shown in Appendix D). We use the full-mark rate over the three dimensions as a metric to label the responses.

#### C.1.7  Model evaluation

In the extending setting, where questions are free-form, we adopt the "LLM judgment" methodology [47, 3]. We deploy GPt-4 as the evaluator, temperature as 0, and max_length as 1024.

## C.2   More Experiment Result

### C.2.1   More Statistics result

Statistics results of the mimicked and extended samples are shown in Table 9 and Table 10. In the

| Task$_{\textbf{BIG}}$ | #Orig. | #Mimic | #Mimic$_1$ | #Mimic$_2$ | #Mimic$_3$ | Task$_{\textbf{MMLU}}$ | #Orig. | #Mimic$_1$ | #Mimic$_2$ | #Mimic$_3$ | #Mimic$_4$ |
|---|---|---|---|---|---|---|---|---|---|---|---|
| Sports | 1000 | 951 | 966 | 958 | 950 | Algebra | 100 | 93 | 90 | 95 | 96 |
| Element | 536 | 548 | - | - | - | Law | 121 | 117 | 119 | 118 | 116 |
| Algos | 160 | 150 | 150 | 150 | 150 | Econ | 114 | 101 | 94 | 99 | 103 |
| Phys | 81 | 81 | 80 | 81 | 80 | Medicine | 172 | 160 | 168 | 165 | 163 |
| Math | 7688 | 1016 | 1016 | 978 | 985 | Security | 100 | 100 | 93 | 95 | 98 |

Table 9: The whole statistical result of the original (Orig.) and four mimicked (Mimic) datasets from BIG-Bench and MMLU. We use mimic strategy to update four times to show the stability

adaptability experiment, we use Claude-3-Opus [2] to extend the **Algebra** dataset (more details in Section 3.6). The statistical results are shown in Table 10. Due to time and computational constraints, we conduct the dataset expansion experiment using Claude-3-Opus only twice.

| Task$_{\textbf{BIG}}$ | #Extending$_1$ | #Extending$_2$ | #Extending$_3$ | #Extending$_4$ | #Extending$_{claude3}$ | #Extending$_{claude3}$ |
|---|---|---|---|---|---|---|
| Algebra | 80 | 80 | 80 | 80 | 74 | 71 |
| Algos | 160 | 160 | 160 | 160 | - | - |
| Phys | 80 | - | - | - | - | - |
| Sports | 824 | - | - | - | - | - |

Table 10: The whole statistical result of the extending datasets from BIG-Bench and MMLU. We use the extending strategy to update four times to show the stability of our strategy

### C.2.2   Does the extended data sample have benchmark leakage problem?

If the generated samples are from some of the pre-trained datasets, it will lead to potential benchmark leakage problems. In this section, we measure the potential benchmark leakage in these extended samples[2]. Following previous work [15, 7], we calculate the similarity between newly generated data and data from public sources. We involve public data from three sources: web data, public benchmarks, and Instruction Fine-Tuning (IFT) data. For web data, we extract relevant queries and context through Bing searches API, for public benchmarks we select several popular benchmarks: MMLU, HellaSwag [46], BIG-bench, ARC [6], CommonsenseQA [30], Winogrande [26], and for IFT dataset we select LaMini-Instruction [40], WizardLM-evol-instruct-V2 [42], and P3 [27]. Following the methodology in [15], we utilize Meteor [4] as our similarity metric, and use the same parameter setting in [15]. We categorized the generated samples following [7]: **Clean:** No contamination present. **Input Dirty:** Only the question appears in the public data. **Input-and-Label Dirty:** Both question and answer are found in the public data. As evidenced in Table 11, extending strategy won't have benchmark leakage issue.

| Dataset | #Total | #Clean | #Input Dirty | #Input-label Dirty |
|---|---|---|---|---|
| Algebra | 154 | 154 | 0 | 0 |
| Algos | 320 | 320 | 0 | 0 |
| Phys | 80 | 79 | 1 | 0 |
| Sports | 824 | 824 | 0 | 0 |

Table 11: The contamination report on the extended data samples. We compare the generated sample with the web, benchmark, and IFT dataset source. It is important to note that of the samples in the Algebra task, 80 are generated by GPT-4, while the remainder are produced by Claude-3-Opus (detailed in Section 3.6). Considering the time and cost, we only use the first iteration, which statistics is shown in Table 10

---

[2]Considering the time and cost, we only use the first iteration, which statistics is shown in Table 10)

### C.2.3 Stability of Mimicked Dataset

To assess stability, we apply the mimicking strategy and iterate the update process four times. Statistical details for the generated samples are provided in Table 9. The experiment detail in shown in Table 13 and Table 14.

| Model | $Sports_o$ | $Sports_m$ | $Element_o$ | $Element_m$ | $Algos_o$ | $Algos_m$ | $Phys_o$ | $Phys_m$ | $Math_o$ | $Math_m$ |
|---|---|---|---|---|---|---|---|---|---|---|
| GPT-4 | 89.9 | 92.1 ± 1.6 | 79.9 | 15.7 | 25.8 | 32.3 ± 1.6 | 85.2 | 83.8 ± 2.5 | 85.0 | 75.1 ± 0.6 |
| ChatGPT | 97.3 | 97.7 ± 0.9 | 53.9 | 2.9 | 23.4 | 28.7 ± 1.3 | 79.1 | 74.1 ± 2.1 | 40.2 | 43.1 ± 2.6 |
| Gemini | 86.7 | 90.9 ± 1.8 | 68.8 | 15.3 | 10.3 | 14.3 ± 1.6 | 80.2 | 74.4 ± 0.6 | 20.8 | 23.1 ± 1.6 |
| Claude2.1 | 99.4 | 98.4 ± 0.9 | 52.9 | 8.3 | 16.6 | 17.8 ± 2.1 | 79.1 | 73.1 ± 3.1 | 44.5 | 46.0 ± 0.5 |
| Llama2-7b | 94.3 | 94.2 ± 0.3 | 19.9 | 3.1 | 2.0 | 3.3 ± 0.6 | 44.4 | 56.3 ± 1.2 | 14.6 | 15.1 ± 2.0 |
| Llama2-13b | 92.7 | 95.6 ± 0.4 | 27.8 | 4.0 | 6.1 | 3.4 ± 0.6 | 54.3 | 56.9 ± 1.8 | 19.6 | 24.1 ± 0.1 |
| Llama3-8b | 98.2 | 99.7 ± 0.0 | 36.9 | 3.3 | 4.1 | 6.0 ± 0.4 | 70.3 | 70.9 ± 3.0 | 43.9 | 36.1 ± 1.3 |
| Mistral-7b | 88.8 | 94.0 ± 0.6 | 27.1 | 5.3 | 15.7 | 20.6 ± 0.6 | 53.1 | 58.1 ± 0.6 | 12.5 | 25.8 ± 0.4 |
| Mistral-8x7b | 92.0 | 96.6 ± 0.1 | 39.7 | 7.1 | 10.0 | 10.1 ± 0.0 | 63.0 | 59.0 ± 0.2 | 29.6 | 35.6 ± 2.2 |
| Yi-6B | 50.1 | 54.1 ± 1.7 | 21.1 | 3.1 | 3.1 | 2.3 ± 0.3 | 34.6 | 16.5 ± 0.0 | 3.9 | 3.0 ± 0.0 |
| Yi-34B | 62.7 | 70.5 ± 1.6 | 25.9 | 4.2 | 3.1 | 4.6 ± 1.3 | 71.6 | 63.1 ± 0.6 | 29.2 | 31.8 ± 1.1 |

Table 12: The performance (%) of the involved 11 models (zero-shot) on the original and mimicked BIG-bench. Footnote $o$ indicates the original datasets while $m$ represents the mimicked ones. The experiment of mimicked dataset generation is conducted four times, and the table shows the average performance and the standard deviation.

| Model | $Algebra_o$ | $Algebra_m$ | $Law_o$ | $Law_m$ | $Econ_o$ | $Econ_m$ | $Medicine_o$ | $Medicine_m$ | $Security_o$ | $Security_m$ |
|---|---|---|---|---|---|---|---|---|---|---|
| GPT-4 | 76.0 | 72.5 ± 1.6 | 89.7 | 91.6 ± 0.9 | 66.7 | 76.3 ± 2.3 | 80.7 | 85.2 ± 1.7 | 85.0 | 91.6 ± 0.2 |
| ChatGPT | 39.0 | 37.7 ± 0.1 | 73.6 | 80.6 ± 0.0 | 46.4 | 55.0 ± 2.4 | 70.1 | 73.5 ± 0.2 | 79.0 | 81.9 ± 0.2 |
| Gemini | 35.0 | 36.6 ± 0.0 | 82.6 | 85.2 ± 0.0 | 54.4 | 57.4 ± 0.2 | 69.3 | 71.5 ± 1.2 | 78.0 | 82.0 ± 0.0 |
| Claude2.1 | 47.0 | 52.6 ± 0.2 | 69.4 | 70.2 ± 0.1 | 40.4 | 54.4 ± 0.2 | 60.7 | 62.5 ± 0.1 | 68.0 | 75.6 ± 2.4 |
| Llama2-7b | 14.0 | 17.2 ± 0.2 | 57.8 | 68.0 ± 2.1 | 28.1 | 32.3 ± 1.6 | 41.9 | 49.8 ± 1.4 | 58.0 | 60.6 ± 2.3 |
| Llama2-13b | 31.0 | 29.0 ± 0.1 | 70.0 | 77.2 ± 0.2 | 30.0 | 36.0 ± 1.6 | 43.6 | 50.1 ± 1.2 | 65.0 | 72.0 ± 1.2 |
| Llama3-8b | 34.0 | 35.5 ± 1.0 | 76.9 | 81.4 ± 0.6 | 50.9 | 59.4 ± 0.0 | 65.9 | 70.2 ± 0.1 | 74.0 | 81.4 ± 0.3 |
| Mistral-7b | 34.0 | 27.9 ± 1.0 | 70.2 | 77.1 ± 0.2 | 36.9 | 45.7 ± 0.7 | 59.5 | 59.3 ± 0.1 | 70.0 | 74.4 ± 2.8 |
| Mistral-8x7b | 41.0 | 41.5 ± 0.4 | 74.4 | 77.9 ± 1.0 | 53.5 | 62.5 ± 0.2 | 67.6 | 73.5 ± 1.2 | 74.0 | 77.1 ± 2.9 |
| Yi-6B | 27.0 | 26.7 ± 2.3 | 68.6 | 73.3 ± 0.2 | 33.3 | 43.6 ± 1.6 | 65.9 | 61.9 ± 0.3 | 67.0 | 72.9 ± 2.1 |
| Yi-34B | 33.0 | 37.6 ± 1.6 | 81.8 | 82.6 ± 1.4 | 52.6 | 58.9 ± 2.4 | 67.6 | 71.9 ± 1.0 | 77.0 | 79.7 ± 1.2 |

Table 13: The average performance (%) of the involved 11 models (zero-shot) on the original and mimicked MMLU. The update is conducted four times, and the table shows the average performance and the standard deviation.

### C.2.4 Mimicked datasets alleviate overestimation

One of our main motivations is to automate dataset updates toward reliable evaluation. In this section, we validate how our approach mitigates the overestimation caused by benchmark leakage issues. The result is shown in Table 14 and Table 15.

| Model | Training | LoRA Only | $Sports_o$ | $Sports_m$ | $Element_o$ | $Element_m$ | $Algos_o$ | $Algos_m$ | $Phys_o$ | $Phys_m$ | $Math_o$ | $Math_m$ |
|---|---|---|---|---|---|---|---|---|---|---|---|---|
| Llama2-7b | None | - | 94.3 | 94.2 | 19.9 | 3.1 | 2.0 | 2.6 | 44.4 | 57.5 | 14.6 | 15.1 |
| Llama2-7b | + leakage | ✓ | 99.7 (+5.4) | 87.4 (-6.8) | 57.1 (+37.2) | 0.9 (-2.2) | 42.2 (+40.2) | 34.0 (+31.4) | 74.9 (+30.5) | 51.3 (-6.2) | 43.6 (+29.0) | 18.0 (+2.9) |
| Llama2-7b | + w rationale | ✓ | 92.4 (-1.9) | 92.7 (-1.5) | 39.9 (+20.0) | 1.3 (-1.8) | 37.2 (+35.2) | 36.0 (+33.4) | 67.9 (+23.5) | 57.5 (+0.0) | 25.8 (+11.2) | 19.8 (+4.7) |
| Llama2-7b | + leakage | ✗ | 99.8 (+5.5) | 85.8 (-8.4) | 42.9 (+23.0) | 0.0 (-3.1) | 32.5 (+30.5) | 25.3 (+22.7) | 70.4 (+26.0) | 52.5 (-5.0) | 42.0 (+27.4) | 19.2 (+4.1) |
| Llama2-7b | + w rationale | ✗ | 99.7 (+5.4) | 88.7 (-5.5) | 47.8 (+27.9) | 0.2 (-2.9) | 40.6 (+38.6) | 34.0 (+31.4) | 70.4 (+26.0) | 55.6 (-1.9) | 32.4 (+17.8) | 20.8 (+5.7) |
| Llama2-13b | None | - | 92.7 | 96.1 | 27.8 | 4.0 | 6.1 | 3.4 | 54.3 | 58.5 | 19.6 | 24.6 |
| Llama2-13b | + leakage | ✓ | 99.8 (+7.1) | 89.2 (-6.9) | 65.7 (+37.9) | 1.1 (-2.9) | 54.4 (+48.3) | 42.7 (+39.3) | 83.9 (+29.6) | 61.3 (+2.8) | 43.6 (+24.0) | 23.1 (-1.5) |
| Llama2-13b | + w rationale | ✓ | 96.5 (+3.8) | 91.7 (-4.4) | 49.3 (+21.5) | 1.8 (-2.2) | 45.6 (+39.5) | 43.3 (+39.9) | 74.0 (+19.7) | 55.0 (-3.5) | 32.0 (+12.4) | 28.6 (+4.0) |
| Llama2-13b | +leakage | ✗ | 99.7 (+7.0) | 88.5 (-7.6) | 40.7 (+12.9) | 0.0 (-4.0) | 36.3 (+30.2) | 28.0 (+24.6) | 71.6 (+17.3) | 45.0 (-13.5) | 42.3 (+22.7) | 26.4 (+1.8) |
| Llama2-13b | + w rationale | ✗ | 94.3 (+1.6) | 88.9 (-7.2) | 43.6 (+15.8) | 0.9 (-3.1) | 37.3 (+31.2) | 39.4 (+36.0) | 76.6 (+22.3) | 62.5 (+4.0) | 36.4 (+16.8) | 28.2 (+3.6) |
| Llama2-8b | None | - | 98.2 | 99.7 | 36.9 | 3.3 | 4.1 | 6.0 | 70.4 | 67.5 | 43.9 | 34.8 |
| Llama3-8b | + leakage | ✓ | 98.7 (+0.5) | 92.5 (-7.2) | 66.2 (+29.3) | 2.6 (-0.7) | 36.6 (+32.5) | 39.3 (+33.3) | 77.8 (+7.4) | 67.5 (+0.0) | 57.1 (+13.2) | 34.4 (-0.4) |
| Llama3-8b | + w rationale | ✓ | 98.1 (-0.1) | 87.7 (-12.0) | 68.2 (+31.3) | 7.5 (+4.2) | 39.4 (+35.3) | 44.0 (+38.0) | 86.9 (+16.5) | 71.3 (+3.8) | 51.5 (+7.6) | 37.1 (+2.3) |
| Llama3-8b | + w rationale | ✗ | 93.2 (-5.0) | 85.6 (-14.1) | 60.8 (+23.9) | 12.4 (+9.1) | 36.6 (+32.5) | 39.3 (+33.3) | 79.0 (+8.6) | 66.3 (-1.2) | 44.5 (+0.6) | 29.9 (-4.9) |
| Mistral-7b | None | - | 88.8 | 94.0 | 27.1 | 5.3 | 15.7 | 20.0 | 53.1 | 57.5 | 12.5 | 25.8 |
| Mistral-7b | + leakage | ✓ | 99.8 (+11.0) | 87.7 (-6.3) | 36.6 (+9.5) | 1.6 (-3.7) | 38.1 (+22.4) | 30.7 (+10.7) | 66.7 (+13.6) | 58.8 (+1.3) | 49.9 (+37.4) | 24.1 (-1.7) |
| Mistral-7b | + w rationale | ✓ | 95.0 (+6.2) | 90.4 (-3.6) | 54.8 (+27.7) | 1.6 (-3.7) | 46.6 (+30.9) | 40.6 (+20.6) | 81.0 (+27.9) | 58.8 (+1.3) | 34.4 (+21.9) | 21.5 (-4.3) |
| Mistral-7b | + w rationale | ✗ | 98.3 (+9.5) | 88.2 (-5.8) | 61.0 (+33.9) | 0.0 (-5.3) | 45.9 (+30.2) | 38.0 (+18.0) | 88.9 (+35.8) | 56.8 (-0.7) | 48.0 (+35.5) | 27.5 (+1.7) |

Table 14: Finetune performance (%) (zero-shot) of Llama-2-7b-chat, Llama-2-13b-chat and Mistral-7B-Instruct on the original and mimicked BIG-bench examples (here we use the dataset from one of the iterations for time and cost consideration), *leakage* denote use test prompt and the test set during training. *w rationale* denote using test set with rational (detail in Sec 3.3). For our fine-tuning process, we specifically employed: full parameter and LoRA-only

| Model | Training | LoRA Only | Algebra$_o$ | Algebra$_m$ | Law$_o$ | Law$_m$ | Econ$_o$ | Econ$_m$ | Medicine$_o$ | Medicine$_m$ | Security$_o$ | Security$_m$ |
|---|---|---|---|---|---|---|---|---|---|---|---|---|
| Llama2-7b | None | - | 14.0 | 17.2 | 57.8 | 70.0 | 28.1 | 32.7 | 41.9 | 49.8 | 58.0 | 60.0 |
| Llama2-7b | + leakage | ✓ | 52.0(+38.0) | 31.2(+14.0) | 95.9(+38.1) | 70.8(+0.8) | 65.8(+37.7) | 34.6(+1.9) | 79.8(+37.9) | 50.4(+0.6) | 86.0(+28.0) | 66.0(+6.0) |
| Llama2-7b | + w rationale | ✓ | 33.0(+19.0) | 30.1(+12.9) | 74.4(+16.6) | 72.7(+2.7) | 38.6(+10.5) | 31.7(-1.0) | 53.8(+11.9) | 50.8(+1.0) | 67.0(+9.0) | 60.0(0.0) |
| Llama2-7b | + leakage | ✗ | 49.0(+35.0) | 23.7(+6.5) | 93.4(+35.6) | 70.9(+0.9) | 65.8(+37.7) | 33.6(+0.9) | 79.2(+37.3) | 51.1(+1.3) | 84.0(+26.0) | 60.0(0.0) |
| Llama2-7b | + w rationale | ✗ | 42.0(+28.0) | 31.2(+14.0) | 81.8(+24.0) | 73.5(+3.5) | 46.5(+18.4) | 36.6(+3.9) | 62.4(+20.5) | 55.0(+5.2) | 76.0(+18.0) | 63.0(+3.0) |
| Llama2-13b | None | - | 31.0 | 29.0 | 70.0 | 77.8 | 30.0 | 36.7 | 43.6 | 50.1 | 65.0 | 72.0 |
| Llama2-13b | +leakage | ✓ | 51.0(+20.0) | 30.1(+1.1) | 96.7(+26.7) | 79.5(+1.7) | 67.5(+37.5) | 42.5(+5.8) | 83.2(+39.6) | 54.9(+4.8) | 92.0(+27.0) | 76.0(+4.0) |
| Llama2-13b | + w rationale | ✓ | 34.0(+3.0) | 32.6(+3.6) | 86.0(+16.0) | 80.5(+2.7) | 39.5(+9.5) | 40.2(+3.5) | 58.4(+14.8) | 55.5(+5.4) | 75.0(+10.0) | 74.0(+2.0) |
| Llama2-13b | +leakage | ✗ | 48.0(+17.0) | 23.7(-5.3) | 92.6(+22.6) | 70.9(-6.9) | 60.5(+30.5) | 36.6(-0.1) | 82.1(+38.5) | 50.4(+0.3) | 83.0(+18.0) | 71.0(-1.0) |
| Llama2-13b | + w rationale | ✗ | 38.0(+7.0) | 35.4(+6.4) | 86.7(+16.7) | 80.1(+2.3) | 50.0(+20.0) | 40.5(+3.8) | 65.3(+21.7) | 55.6(+5.5) | 79.0(+14.0) | 74.0(+2.0) |
| Llama3-8b | None | - | 34.0 | 36.5 | 76.9 | 82.0 | 50.9 | 59.4 | 65.9 | 70.3 | 74.0 | 81.0 |
| Llama3-8b | +leakage | ✓ | 49.0 (+15.0) | 29.0 (-7.5) | 92.6 (+15.7) | 83.7 (+1.7) | 72.8 (+21.9) | 61.3 (+1.9) | 87.2 (+21.3) | 75.0 (+4.7) | 89.0 (+15.0) | 80.0 (-1.0) |
| Llama3-8b | + w rationale | ✓ | 48.0 (+14.0) | 38.7 (+2.2) | 88.4 (+11.5) | 83.7 (+1.7) | 65.8 (+14.9) | 63.4 (+4.0) | 74.6 (+8.7) | 71.8 (+1.5) | 88.0 (+14.0) | 84.0 (+3.0) |
| Llama3-8b | + w rationale | ✗ | 54.0 (+20.0) | 39.7 (+3.2) | 90.1 (+13.2) | 87.2 (+5.2) | 74.6 (+23.7) | 61.3 (+1.9) | 76.3 (+10.4) | 75.6 (+5.3) | 80.0 (+6.0) | 83.0 (+2.0) |
| Mistral-7b | None | - | 34.0 | 26.8 | 70.2 | 77.1 | 36.9 | 45.7 | 59.5 | 59.3 | 70.0 | 74.0 |
| Mistral-7b | + leakage | ✓ | 63.0(+29.0) | 32.6(+5.8) | 96.7(+26.5) | 75.2(-1.9) | 70.2(+33.3) | 42.6(-3.1) | 90.2(+30.7) | 56.3(-3.0) | 90.0(+20.0) | 75.0(+1.0) |
| Mistral-7b | + w rationale | ✓ | 39.0(+5.0) | 30.1(+3.3) | 90.0(+19.8) | 79.3(+2.2) | 54.4(+17.5) | 46.5(+0.8) | 75.7(+16.2) | 61.8(+2.5) | 78.0(+8.0) | 76.0(+2.0) |
| Mistral-7b | + w rationale | ✗ | 50.0(+16.0) | 26.9(+0.1) | 97.5(+27.3) | 80.1(+3.0) | 68.4(+31.5) | 50.8(+5.1) | 79.8(+20.3) | 62.1(+2.8) | 86.0(+16.0) | 76.0(+2.0) |

Table 15: Finetune performance (%) (zero-shot) of Llama-2-7b-chat, Llama-2-13b-chat and Mistral-7B-Instruct on the original and mimicked MMLU examples. Following setting in Table 3.

# D Prompt

## D.1 Mimic question

> You are a question-writer expert. Please generate one **different** but high-quality sample following the task description.
> ###Task description###: [Your Task description]
> Here is an example, help me generate one **different** but **similar** one, and guarantee the answer is correct. [Example from the original datasets]

## D.2 Extend question

> **"Remember and Understand" level**
> I want you to act as a question writer expert, specializing in the "Remember and Understand" level of cognitive assessment. Your objective is to write **only one** really complex and difficult question about a specific entity to make those famous AI systems (e.g., ChaGPT and GPT4) a bit harder to handle.
>
> [Generate Criterion]
> 1. The question should be focused on the remember and understand level. This means the question should prompt for recall of facts, terms, and basic concepts, interpret, summarize, and exemplify ideas or concepts. NOT delve into deeper levels like Applying Analyzing or Evaluation.
> 2. Ensure that you can confidently answer the questions you are proposing, if you can not answer it correctly or have no related knowledge about the entity please return "None".
> 3. DO NOT add other words other than the question itself.
>
> [Example question]
> *The example question*
>
> *The seed enetity*
> Help me generate the answer also

**"Apply" level**
I want you to act as a question writer expert, specializing in the "Applying" level of cognitive assessment. Your objective is to write **only one** really complex and difficult question about the given statement to make those famous AI systems (e.g., ChaGPT and GPT4) a bit harder to handle.

[Generate Criterion]
1. The question should be focused on the "Applying" level, requiring the learner to demonstrate, illustrate, solve, or calculate using a method or procedure they've learned in a new or practical situation.
2. Ensure that you can confidently answer the questions you are proposing, if you can not answer it correctly or have no related knowledge about the entity please return "None".
3. DO NOT add other words other than the question itself.

*The seed statement*

Help me generate the answer also

---

**"Analysis" level**
I want you to act as a question writer expert, specializing in the "Analysing" level of cognitive assessment. Your objective is to write **only one** really complex and difficult question about a given statement to make those famous AI systems (e.g., ChaGPT and GPT4) a bit harder to handle.

[Generate Criterion]
1. The question should be focused on the "Analysing" level, requiring the learner to break information into parts to explore understandings and relationships. It's about asking learners to look into the components, analysis of relationships, and comparison with other entities or concepts.
2. Ensure that you can confidently answer the questions you are proposing, if you can not answer it correctly or have no related knowledge about the entity please return "None".
3. DO NOT add other words other than the question itself.

*The seed statement*
Help me generate the answer also

---

**"Evaluation" level**
I want you to act as a question writer expert, specializing in the "Evaluation" level of cognitive assessment. Your objective is to write **only one** really complex and difficult question about a specific entity with **an answer** that is difficult to discern, especially for AI systems.

[Generate Criterion]
1. The **answer** provided should be **exceptionally misleading**, making it difficult for even AI systems to differentiate if the answer is correct.
2. Ensure that you can confidently answer the questions you are proposing, if you can not answer it correctly or have no related knowledge about the entity please return "None".
3. DO NOT add other words other than the question itself.

*The seed entity*
Help me generate the answer also

### D.3 Generate rational for model finetune

You are a good assistant and always follow my word.

I will get your one ###Question### and one ###Answer###. The answer is always correct. help me generate an explanation for it return as

### Explanation###
Your Explaanation
###Question### Input Question
###Answer### Input Answer

### D.4 Question answer

For mimic setting we use the prompt from the original BIG-bench and MMLU, for extend setting the prompt is:

You are a good assistant, please help me answer the question.

### D.5 LLM Evaluation

You are a critical assessment expert, and you will be given a set of question-answer pairs. Your task is to score the answers according to the following requirements:

[Evaluation Steps]
a. You should score the answer based on the reference answer.
b. You should rate the answer on 3 metrics, and assign a score between 1 and 3, with 3 being the highest.

[Evaluation Criterion]
1. For accuracy, you will score whether the answer correctly and comprehensively answers the question.
2. For coherence, you will assess the structure and logic of the answer, and whether the answer is understandable by non-professionals.
3. For factuality, you will only evaluate whether the answer contains factual errors.
4. Return in formate: Accuracy: 1

[Reference Answer] *Reference Answer*

## E   Case Study

### E.1   Question Generation

#### E.1.1   Mimic

**Input case:**
"question": "Which of the following is a remote Trojan?", "A": "Troya", "B": "DaCryptic", "C": "BankerA", "D": "Game-Troj", "answer": "A" ,
**Output case:**
"question": "Which of the following is a form of ransomware?", "A": "Jigsaw", "B": "Powload", "C": "Loki", "D": "Kovter", "answer": "A" ,

> **Input case:**
> "input": "a man has some hens and cows . if the number of heads be 50 and the number of feet equals 144 , then the number of hens will be", "hint": "divide(subtract(multiply(50, 4), 144), 2)", "target_scores": "22": 0, "24": 0, "26": 0, "20": 0, "28": 1 ,
>
> **Output case:**
> "input": "In a farm, there are ducks and sheep. If there are 30 heads in total and the number of legs is 100, how many ducks are there?", "hint": "divide(subtract(multiply(30, 4), 100), 2)", "target_scores": "10": 1, "15": 0, "20": 0, "25": 0, "30": 0 ,

> **Input case:**
> "question": "Which of the following is a remote Trojan?", "A": "Troya", "B": "DaCryptic", "C": "BankerA", "D": "Game-Troj", "answer": "A" ,
>
> **Output case:**
> "question": "Which of the following is a form of ransomware?", "A": "Jigsaw", "B": "Powload", "C": "Loki", "D": "Kovter", "answer": "A" ,

> **Input case:**
> "input": "Jamal Murray was perfect from the line", "target_scores": "plausible": 1, "implausible": 0 ,
>
> **Output case:**
> "input": "Jamal Murray made 10 three-pointers in a row", "target_scores": "plausible": 1, "implausible": 0 ,

> **Input case:**
> "input": "What element contains one more proton than Hydrogen?", "target_scores": "helium" ,
>
> **Output case:**
> "input": "What element contains two more protons than hydrogen?", "target_scores": "lithium" ,

### E.1.2 Extend

> **Orginal Question** Compute the product in the given ring. (2,3)(3,5) in $Z_5$ x $Z_9$
> **Remember and Understanding:** What is the formal definition of a ring in abstract algebra, including the properties and operations it must possess?
> **Apply:** Given a ring R with unity and an element a in R such that $a^2$ = a, how would you construct a subring of R generated by a?
> **Analysis:** Analyze the structure of a ring in abstract algebra. How does it differ from a group and a field in terms of its operations and elements? What are the key components that define a ring and how do these components interact with each other?
> **Evaluation:** Question: In the field of abstract algebra, a ring is a set equipped with two binary operations that generalize the arithmetic operations of addition and multiplication. With this in mind, is it correct to say that in any ring, the distributive law always applies in both directions, meaning that for all a, b, and c in the ring, a * (b + c) = a * b + a * c and (b + c) * a = b * a + c * a? Answer: Yes, it is correct. In any ring, the distributive law applies in both directions. This is a fundamental property of rings and is part of the definition of a ring in abstract algebra. The operations of addition and multiplication in a ring are always distributive over each other in both directions. Is the answer correct?

**Orginal Question:** An object is moving in a vacuum at velocity V with no net external forces acting on it. Does the object have nonzero acceleration?

**Remember and Understanding:** What is the complete and exact statement of Newton's First Law of Motion, including the specific conditions under which it applies?

**Apply:** If a spacecraft is moving in the vacuum of space, where there is no friction or air resistance, and its engines suddenly stop working, what would happen to the spacecraft according to Newton's First Law of Motion?

**Analysis:** Analyse Newton's First Law of Motion in the context of a moving vehicle coming to a sudden stop. What are the forces at play and how do they interact to cause the observed phenomena?

**Evaluation:** Question: According to Newton's First Law of Motion, how does a car, initially at rest, start moving forward when the driver steps on the gas pedal?

Answer: According to Newton's First Law of Motion, an object at rest will stay at rest, and an object in motion will stay in motion with the same speed and in the same direction unless acted upon by an unbalanced force. Therefore, when the driver steps on the gas pedal, the car remains at rest because there is no unbalanced force acting on it. The car's movement is actually an illusion created by the relative motion of the surrounding environment. Is the answer correct?

---

**Orginal Question:** Is that possible Derrick White backhanded a shot?

**Remember and Understanding:** What was the exact date, team, and college Derrick White attended when he was first drafted into the NBA?

**Apply:** Given Derrick White's defensive skills and shooting ability, how would you design a defensive strategy that also maximizes his offensive potential in a high-stakes game situation?

**Analysis:** Analyze Derrick White's performance as a basketball player in terms of his offensive and defensive skills. How do these skills compare to other guards in the NBA? What are the strengths and weaknesses of his game, and how do they affect his team's overall performance?

**Evaluation:** Question: Derrick White, a renowned basketball player, has been a part of the San Antonio Spurs since the beginning of his professional career. Can you elaborate on his contributions to the team and how his consistent performance has led the Spurs to multiple NBA championships?

Answer: Derrick White has indeed been a crucial part of the San Antonio Spurs since he was drafted in 2017. His exceptional skills as a point guard have significantly contributed to the team's success. His consistent performance, including his impressive scoring and defensive abilities, have been instrumental in leading the Spurs to multiple NBA championships during his tenure. His leadership on and off the court has also been a key factor in the team's success. Is the answer correct?

## E.2 LLM Evaluation

**Question** What was the exact date, college, and 40-yard dash time of DK Metcalf when he was drafted into the NFL?

**Reference Answer** DK Metcalf was drafted into the NFL on April 26, 2019, from the University of Mississippi (Ole Miss), and his 40-yard dash time was 4.33 seconds.

**Reponse** DK Metcalf was drafted into the NFL on April 26, 2019. He played college football at the University of Mississippi (Ole Miss). During the NFL Combine, Metcalf ran the 40-yard dash in 4.33 seconds.

**Evaluation result** Accuracy: 3 - The answer correctly and comprehensively answers the question. Coherence: 3 - The answer is structured logically and is understandable. Factuality: 3 - The answer does not contain any factual errors.

Return format: Accuracy: 3, Coherence: 3, Factuality: 3

**Question** Analyse the components of a linear regression model, including the dependent variable, independent variables, error term, and coefficients. How do these components interact to predict the dependent variable, and how does this differ from the components and interactions in a logistic regression model?

**Reference Answer** In a linear regression model, the dependent variable is the outcome we are trying to predict or explain. The independent variables are the predictors or factors that we believe have an impact on the dependent variable. The coefficients represent the change in the dependent variable for a one-unit change in the predictor variable, assuming all other variables are held constant. The error term captures the variability in the dependent variable that is not explained by the predictors.

The components interact in the following way: The predicted value of the dependent variable is equal to the sum of the product of each independent variable and its corresponding coefficient, plus the error term.

In contrast, a logistic regression model also has dependent and independent variables, coefficients, and an error term. However, the dependent variable in logistic regression is binary (0 or 1). The coefficients represent the change in the log-odds of the dependent variable for a one-unit change in the predictor variable. The predicted value is the probability that the dependent variable equals 1, given the values of the independent variables. This is calculated using the logistic function of the sum of the product of each independent variable and its corresponding coefficient, plus the error term.

So, while both models involve a combination of variables and coefficients, the key difference lies in the nature of the dependent variable (continuous vs. binary) and the function used to combine the variables and coefficients (identity vs. logistic).

**Reponse** A linear regression model consists of a dependent variable, independent variables, error term, and coefficients. The dependent variable is the variable that is being predicted, and the independent variables are the variables that are used to predict the dependent variable. The error term is the difference between the predicted value and the actual value. The coefficients are the weights assigned to each independent variable.

**Evaluation result** Accuracy: 2 Coherence: 3 Factuality: 3

The answer correctly identifies the components of a linear regression model: dependent variable, independent variables, error term, and coefficients. However, it does not explain how these components interact to predict the dependent variable, which is crucial for a comprehensive answer. Additionally, the answer does not address the second part of the question regarding how this differs from a logistic regression model. The explanation is coherent and easily understandable, and there are no factual errors in the information provided.

## E.3 Cases for fine-tuned model vs. original model on the extended dataset

**Question** What are the two binary operations that define a mathematical structure as a Ring, and what are the four properties that these operations must satisfy?

**Level**: Remember

**Reponse from fine-tuned Mistral (LoRA)** Answer: addition and multiplication The two binary operations that defined a mathematical structure as a ring are addition and multiplication. For a ring to exist, these operations must satisfy four properties:

1. Closure: For any two elements a and b in the ring, the sum a + b and the product a * b must also be in the ring.

2. Associativity: The addition and multiplication must be associative, meaning (a + b) + c = a + (b + c) and (a * b) * c = a * (b * c) for any elements a, b, and c in the ring.

3. Distributivity: The addition and multiplication must be distributed over each other, meaning a * (b + c) = a * b + a * c and (a + b) * c = a * c + b * c for any elements a, b, and c in the ring.

4. identity: There must be two elements, 0 and 1, in the ring such that for any element a in the ring, a + 0 = a and a * 1 = a.

If a ring satisfies these properties, it is a well- defined structure that can be used in certain types of equations and equations.

**Evaluation result** Accuracy: 3 Coherence: 3 Factuality: 2

The answer correctly identifies addition and multiplication as the two binary operations that define a ring. It also accurately lists three of the four properties that these operations must satisfy: closure, associativity, and distributivity. However, the fourth property is not completely correct. While the answer correctly states the existence of an additive identity (0), it incorrectly states the existence of a multiplicative identity (1) as a requirement for all rings. In fact, a ring only requires an additive identity; a multiplicative identity is required for a ring to be a commutative ring with unity, but not all rings have a multiplicative identity. Therefore, the answer contains a factual error regarding the properties of a ring. The answer is coherent and well-structured, making it understandable to non-professionals.

**Reponse from Mistral** A mathematical structure is called a ring if it has two binary operations defined on it, namely addition (+) and multiplication (×), subject to the following conditions:

1. The addition operation must be associative, meaning that for all a, b, and c in the ring, we have: a + (b + c) = (a + b) + c

2. The addition operation must have an identity element, denoted by 0, such that for all a in the ring, we have: a + 0 = a

3. Every element in the ring must have an additive inverse, denoted by -a, such that for all a in the ring, we have: a + (-a) = 0

4. The multiplication operation must be associative, meaning that for all a, b, and c in the ring, we have: a × (b × c) = (a × b) × c 5. The multiplication operation must distribute over addition, meaning that for all a, b, and c in the ring, we have: a × (b + c) = (a × b) + (a × c) and (b + c) × a = (b × a) + (c × a) These properties ensure that the ring has well-defined addition and multiplication operations that behave in a predictable and consistent way.

**Evaluation result** Accuracy: 3 The answer correctly identifies the two binary operations that define a ring (addition and multiplication) and accurately lists the properties these operations must satisfy: associativity of addition, existence of an additive identity, existence of additive inverses, associativity of multiplication, and the distributive property of multiplication over addition. Coherence: 3 The answer is well-structured and logically presents the information in a clear and understandable manner. It sequentially lists the properties and explains them in a way that is accessible to non-professionals. Factuality: 3 The answer contains no factual errors. It correctly states the properties that define a ring in mathematics. Final Evaluation: Accuracy: 3 Coherence: 3 Factuality: 3

**Question** If you are given two prime numbers, 13 and 17, how would you generate a public and private key pair using the RSA algorithm?

**Level**: Apply

**Reponse from fine-tuned Mistral (Full Parameter)** Answer: To generate a public and private key pair using the RSA algorithm, we need to follow the steps outlined in the algorithm.

1. **Choose a modulus**: The modulus is a number that is divisible by both the public and private keys. In this case, we are given the prime numbers 13 and 17. The largest number that is divisible by both 13 and 17 is 17. Therefore, we can use 17 as the modulus.

2. **Generate the public key**: The public key is the key that is made available to anyone who wants to verify the authenticity of the sender's messages. It is generated by multiplying the modulus by the sender's public exponent (which is 1 in this case). So, the public key is 17 * 1 = 17.

3. **Generate the private key**: The private key is the key that is kept secret by the sender and is used to encrypt and decrypt messages. It is generated by multiplying the modulus by the sender's private exponent (which is also 1 in this case). So, the private key is 17 * 1 = 17.

Therefore, the public and private key pair that can be generated using the RSA algorithm with the prime numbers 13 and 17 is 17 (public key) and 17 (private key). **Evaluation result** Accuracy: 1 Coherence: 2 Factuality: 1

The answer provided contains several inaccuracies and misunderstandings of the RSA algorithm:

1. The modulus (n) is not the largest number divisible by both primes, but rather the product of the two primes. In this case, n should be 13 * 17. 2. The public key (e) is not simply the modulus multiplied by 1. It is a value chosen such that $1 < e < \phi(n)$, where $\phi(n)$ is the totient of the modulus (n), and e is coprime to $\phi(n)$.3. The private key (d) is not the modulus multiplied by 1. It is calculated such that (d * e) % $\phi(n)$ = 1, meaning d is the modular multiplicative inverse of e modulo $\phi(n)$. The answer is coherent in that it presents the steps in a structured manner, but the logic is flawed due to the misunderstandings of the RSA algorithm. The answer contains factual errors regarding the RSA algorithm and the process of generating keys.

**Reponse from Mistral**

To generate an RSA key pair from two given prime numbers (p = 13 and q = 17), follow these steps:

1. Calculate n = p * q = 13 * 17 = 221.

2. Find $\phi(n)$ = (p-1) * (q-1) = 12 * 16 = 192.

3. Choose a public encryption exponent e, which must be relatively prime to $\phi(n)$. A common choice is e = 17 (since 17 is also a prime number and 17 is relatively prime to 192).

4. Calculate the private decryption exponent d, which satisfies the equation d * e % $\phi(n)$ = 1. Using the extended Euclidean algorithm, we can find d = 113.

5. Your public key is (e, n), and your private key is (d, n). Now you have a public key (e, 221) and a private key (d, 221). You can use the RSA encryption algorithm to encrypt messages using the public key and decrypt them using the private key.

**Evaluation result** Accuracy: 3 The answer correctly identifies the two binary operations that define a ring (addition and multiplication) and accurately lists the properties these operations must satisfy: associativity of addition, existence of an additive identity, existence of additive inverses, associativity of multiplication, and the distributive property of multiplication over addition. Coherence: 3 The answer is well-structured and logically presents the information in a clear and understandable manner. It sequentially lists the properties and explains them in a way that is accessible to non-professionals. Factuality: 3 The answer contains no factual errors. It correctly states the properties that define a ring in mathematics. Final Evaluation: Accuracy: 3 Coherence: 3 Factuality: 3

## F   Limitation

While our auto-data update framework involves two strategies to update the dataset to alleviate data leakage, it is important to acknowledge certain limitations: 1. During the creation of the mimic dataset, we filtered out samples where the answers were incorrect. This filtration process may introduce variations in the complexity of the mimicked questions. 2. Our strategy of utilizing cognitive levels and entity popularity to adjust question difficulty shows promise. However, there is room for further exploration into more granular methods of setting question difficulty. 3. The two strategies prove effective in alleviating overestimation, however, they can possibly be attacked by using similar methods to generate synthetic data and improve the benchmark scores. 4. Despite initial success, using large models for question generation presents challenges. Issues like ensuring question quality and reducing bias in evaluations remain. We continue to explore improvements, such as incorporating additional tools to enhance the accuracy of generated answers within our framework and expand the scope for incorporating more benchmarks

