# OpenReview forum: "Automating Dataset Updates Towards Reliable and Timely Evaluation of Large Language Models"
_NeurIPS.cc/2024/Datasets_and_Benchmarks_Track — NeurIPS 2024 Track Datasets and Benchmarks Poster_

### Official Review · Reviewer_ajgi · 2024-07-23
**Feedback to 774**

**Rating:** 6
**Confidence:** 4
**Correctness:** The claim is sounded and the evaluati…
**Clarity:** Yes, the paper is well-written

**Review:**

This paper presents a novel and valuable approach to addressing the challenge of constantly updating datasets for evaluating large language models (LLMs). The authors propose two automated dataset update strategies: mimicking and extending. The mimicking strategy generates similar samples based on the original data, preserving stylistic and contextual essence, while the extending strategy expands existing samples at varying cognitive levels. The paper demonstrates the effectiveness of these strategies in mitigating benchmark leakage, controlling difficulty, and ensuring stable evaluation results. The authors' work contributes to the field by providing a novel approach for timely and reliable evaluation of LLMs, enabling fine-grained analysis of their cognitive capabilities.

__Pros__:
* __Novelty__: The proposed dataset update strategies (mimicking and extending) are novel and address a critical challenge in LLM evaluation.
* __Effectiveness__: The paper demonstrates the effectiveness of the proposed strategies in mitigating benchmark leakage, controlling difficulty, and ensuring stable evaluation results.
* __Practicality__: The proposed strategies are practical and can be easily implemented using existing LLM technologies.
* __Relevance__: The paper addresses a highly relevant and timely issue in the field of LLM research.
* __Originality__: The authors propose a novel approach to automating dataset updates for LLM evaluation. This approach is distinct from existing methods that rely on manual curation or static datasets.

__Cons__:
* __limited scope__: The paper focuses on a limited set of benchmarks (BIG-Bench and MMLU). Further evaluation on a wider range of benchmarks would strengthen the paper's claims.
* __Lack of Comparison with Existing Methods__: The paper does not provide a detailed comparison with existing methods for dataset update (even with manually dataset augmentation). Moreover, the selected LLMs to generate new samples are limited. It could introduce biases in the generated datasets due to the LLMs (prompts) used for mimicking and extending. This limitation should be addressed or properly discussed in the paper.
* __Potential for Bias__: The evaluation involves human feedback, this should be discussed (appendix B.1 is not sufficient to be convinced). Human evaluation may affect the accuracy, coherence and clarity of the generated samples and the responses from LLMs. If human evaluation is necessary, the authors should provide more robust evidence to support the validity of the evaluation process. In addition, the evaluation if __Fig. 2/3__ should also include the margins from the original output, assuming you enabled the temperature control/randomness in the selected LLMs.

**Strengths:**

See __pros__ in the review section.

**Additional Feedback:**

N/A

**Documentation:**

The paper provides a website to show the demos of two approaches.

**Ethics:**

Yes, the human evaluations involved in the paper. Potentially, the human evaluation could be biased (data quality and representation) and subjective. The acquired data from human evaluation should be handled with care, and the authors should provide more robust evidence to support the validity of the evaluation process.

**Limitations:**

The author provided limitation section in Appendix F is __incomplete__ (truncated sentences in the end).

**Opportunities For Improvement:**

See __Cons__ in the review section.

**Relation To Prior Work:**

Yes

**Summary And Contributions:**

This paper addresses the challenge of constantly updating datasets for evaluating large language models (LLMs) as they quickly master existing benchmarks. The authors propose two automated dataset update strategies: mimicking and extending. The mimicking strategy generates similar samples based on the original data, preserving stylistic and contextual essence, while the extending strategy expands existing samples at varying cognitive levels. The paper demonstrates the effectiveness of these strategies in mitigating benchmark leakage, controlling difficulty, and ensuring stable evaluation results. The authors' work contributes to the field by providing a novel approach for timely and reliable evaluation of LLMs, enabling fine-grained analysis of their cognitive capabilities.

---

> ### Author Rebuttal · Authors · 2024-08-22
>
> Response to Reviewer ajgi
>
> Thanks for your comprehensive review. Here are our responses on the questions.
>
> - **Regarding "limited set of benchmarks (BIG-Bench and MMLU)"**
>
> **Response 4.1:**  In our research, we utilize two of the most popular and extensive datasets currently available—BIG-Bench and MMLU—for our experiments. The dataset choices are made within the constraints of cost and time, ensuring a comprehensive evaluation. The extensive experiment has shown the reliability of our framework. Thank you for your suggestion. In future work, we plan to expand our dataset scope by incorporating additional benchmarks.
>
> - **Regarding "not provide a detailed comparison with existing methods for dataset update"**
>
> **Response 4.2:** Existing research primarily focuses on detecting data leakage. To our knowledge, our work is the first to alleviate data leakage using automated dataset updates, and as such, there are limited established baselines for a detailed comparison. Despite this, we have thoroughly explored and analyzed the effectiveness and stability of the two strategies we proposed. We plan to expand the related work section of our paper to include a more comprehensive discussion of how our approach connects with and differs from existing work on data leakage.
>
> - **Regarding "It could introduce biases in the generated datasets"**
>
> **Response 4.3:** Thank you for your suggestion. In our previous works, we have proposed a framework that employs multiple large models to mimic peer review processes, aimed at reducing bias in dataset generation and evaluation. Moving forward, our focus indeed is to continue to address and advance this important aspect.
>
>
> - **Regarding the concern "The evaluation involves human feedback"**
>
> **Response 4.4:** Our evaluation pipeline does not use human feedback to generate samples. Our human evaluation process primarily follows traditional guidelines [1]. The human evaluation is to justify if our automated evaluation pipeline can provide an effective evaluation result, as automated assessments often struggle to accurately evaluate the quality of the open-ended text.  Once our automated evaluation pipeline is validated, future updates will not require human evaluation.
>
> - **Regarding your suggestion "the evaluation if Fig. 2/3 should also include the margins from the original output"**
>
> **Response 4.5:** To investigate the consistency of our evaluation results when our mimicking strategy is applied multiple times to the same dataset, we repeat the update process four times. Figures 2 and 3 illustrate the performance on the original dataset (represented by solid bars) also the average performance (represented by dashed bars) and the variation (box plots on the dashed bars) across the four mimicked datasets for the selected model. In the experiment, we intentionally set all the temperature to **0** (Appendix C 1.2 for more detail) to ensure the most stable output from the models — a common setting for model evaluations aiming to minimize variability, ensure reproducibility, as well as accurately reflect the model’s true performance.
>
> [1] DocEE: A Large-Scale and Fine-grained Benchmark for Document-level Event Extraction

---

> > ### Comment · Reviewer_ajgi · 2024-08-27
> >
> > Thank you for providing more details about my concerns. To this point, I will keep my initial rating.

---

> ### Author Response · Authors · 2024-08-26
> **Looking forward to your feedback**
>
> Dear reviewer ajgi,
>
> Thank you again for your helpful reviews. As the end of the discussion period draws near, we kindly request for your feedback and comments. If you find that our responses have effectively addressed your concerns, can you give us feedback? Your feedback is crucial to us. We apologize for any inconvenience caused.

---

### Official Review · Reviewer_La6o · 2024-07-24
**Review for "Automating Dataset Updates Towards Reliable and Timely Evaluation of Large Language Models"**

**Rating:** 6
**Confidence:** 4
**Correctness:** The details are thoroughly mentioned …
**Clarity:** The paper is written very clearly and…

**Review:**

Listed below on each section

**Strengths:**

Dataset contamination is currently one of the most crucial issue for LLM evaluation. The paper attempts to propose an automatic solution to mitigate this issue.

**Additional Feedback:**

None

**Documentation:**

No dataset is explicitly collected, but is automatically created. There are human evaluation results to verify the quality for a subset.

**Limitations:**

1. I think of the biggest limitation is that if GPT-4 is used to generate the answer as well, it means that the resulting dataset couldn't be used to evaluate GPT-4 since it means that it consists of instances that GPT-4 could solve. While utilizing Claude-3 is briefly mentioned in Section 3.6, a more rigorous way (e.g., using an ensemble of data generation models) could also be discussed.

2. The authors argue such that "Our proposed strategies are able to update datasets with minimum human effort whenever needed." (Line 170). However, this is a bit of an over claim considering that a large amount of human prior is still required for each individual task (Line 131 - 138).

**Opportunities For Improvement:**

Also mentioned in the Limitations section, but it would be great if the authors could explain about the dilemma of utilizing powerful proprietary LMs to create better benchmarks. Specifically, since they also generate the label of the instance, it couldn't be used to evaluate that particular model. Yet, it could be still used to evaluate other models in general.

**Relation To Prior Work:**

The authors properly mentions prior works in the related work section.

**Summary And Contributions:**

1. The authors propose a method to automatically bypass the dataset contamination issue by "mimicing" or "extending" an existing benchmark.

2. The authors conduct extensive experiments when adopting their strategy to two representative benchmarks, MMLU and BigBench.

---

> ### Author Rebuttal · Authors · 2024-08-22
>
> Response to Reviewer La6o
>
> Thanks for your review and detailed suggestions. Here are our responses to your questions.
>
> - **Regarding the suggestion "using an ensemble of data generation models could also be discussed."**
>
> **Response 3.1:** Thank you for your suggestion. In our previous works, we have already proposed a framework that employs multiple large models to mimic peer review processes, aimed at reducing bias in dataset generation and evaluation. Moving forward, our focus indeed is to continue to address and advance this important aspect.
>
> - **Regarding the concerns "that a large amount of human prior is still required for each individual task."**
>
> **Response 3.2:** Human prior is not always necessary, as demonstrated by the success of our mimicking strategy in the Sports dataset (Section 3.3). While, for some datasets, introducing human prior can enhance the quality of evaluations, as seen with the extended Sports dataset (Section 3.5).
>
> Here to notice, once this preliminary work is completed, our framework facilitates timely question auto-updates, allowing for the continuous generation of new questions. This capability enables unlimited testing of the model with fresh queries. Particularly with our extension strategy, by simply providing a given seed, we can easily generate multiple test questions that focus on different cognition levels.
>
> In comparison, the effort and cost associated with manually generating corresponding data would be much higher.

---

> ### Author Response · Authors · 2024-08-26
> **Looking forward to your feedback**
>
> Dear reviewer La6o,
>
> Thank you again for your helpful reviews. As the end of the discussion period draws near, we kindly request for your feedback and comments. If you find that our responses have effectively addressed your concerns, can you give us feedback? Your feedback is crucial to us. We apologize for any inconvenience caused.

---

### Official Review · Reviewer_miBT · 2024-07-26

**Rating:** 7
**Confidence:** 3

**Review:**

**Strengths:**
1. The motivation is reasonable and clear, and the automatic updating of the dataset is a valuable research contribution.
2. The experimental results are quite extensive, and the performance has preliminarily met expectations.

**Weaknesses:**
1. Since the datasets used are relatively small (with the mimicked dataset "Math" having only 1000 instances), the generalizability of the proposed method may be questionable.
2. The paper does not discuss the distinctions and connections between the proposed method and existing data leakage work.
3. In Section 2.2, the descriptions of the six cognitive levels in the Extending Strategy are quite abstract. Could a specific example be provided to aid understanding?
4. The term "leakage simulation" mentioned in line 198 is unclear. Please provide a detailed explanation of how it is conducted and how the leakage settings are configured.
5. Models shown in Figures 2 and 3 exhibit significant deviations even in the original data. The author should first explain how these models ensure relatively stable output results. Additionally, based on Figure 3, can it be explained whether the mimicking strategy does not introduce significant differences from the source data? If the models perform well on the source data, they should also perform well on the mimicked data.
6. For example, in Table 4, some tasks show performance degradation with "+leakage" and "+w rationale" settings (e.g., Econ, Medicine), while others show improvement (e.g., Algebra, Security). Is there an explanation for this discrepancy? Currently, it seems more like a random event, and repeating the experiments in Table 4 might show opposite results.
7. Why are there no results for Llama3-8b and Mistral-7b with full parameters and "+leakage" settings? Is it because the results were unsatisfactory?
8. Although the results in Table 5 appear exciting, the paper's premise is updating data. If the data significantly deviates from the task setup, results similar to those in Table 5 could theoretically be achieved. Therefore, the similarity between extended data and public data is crucial. However, I did not find corresponding results in Appendix C.2.2.
9. Why does Section 3.5 not extend the previous setup for experiments on Algebra and Algos but instead focus on Sports and Phys? Is it due to unsatisfactory results?
10. As shown in Figure 5, why is there a significant performance improvement with more popular seeds? A reasonable explanation is needed.
11. Why does Table 6 lack results under the "+leakage" setting? Is it because the results were unsatisfactory?
12. I am curious if extended datasets generated by GPT-4 would still be effective if leakage occurs. Could extended datasets generated by Claude-3-Opus still work to prevent leakage? This would highlight the significance of multi-source extended data.
13. The description of Data Leakage in the Related Work section is somewhat scattered and lacks discussion of how the proposed method differs from and relates to these related works.
14. There are several minor grammatical errors in the paper, such as singular and plural forms (e.g., "leakage issue" to "leakage issues") and the use of definite articles (e.g., "once current" to "once the current"). These need to be corrected.

**Strengths:**

Please refer to the **Review**

**Additional Feedback:**

Please refer to the **Review**

**Clarity:**

The paper performs well in terms of motivation, experimental design, and results. However, it requires further refinement in areas such as dataset scale, comparison with existing work, and experimental setup choices. Please refer to the **Review**

**Correctness:**

The claims in the paper are theoretically sound and supported by experimental results, but the data set construction and benchmark evaluation methods require further refinement and additional details.

**Documentation:**

No

**Limitations:**

The authors have not thoroughly addressed the potential limitations of their work in the paper. For instance, while they propose two strategies for automatically updating datasets—mimicking and extending strategies—they have not adequately discussed the generalizability and reliability of these strategies when applied to larger-scale datasets. To address this, it is recommended that the authors add a section to the paper that specifically examines the performance of these strategies on larger datasets, as well as their generalizability and applicability.

Additionally, the authors should consider and discuss the reasons behind the performance fluctuations observed in the experiments and their possible implications.

**Opportunities For Improvement:**

Please refer to the **Review**

**Relation To Prior Work:**

Almost none

**Summary And Contributions:**

This paper introduces a novel approach to automating updating natural language processing benchmarks. Recognizing that current benchmarks can become mastered or leaked, leading to unreliable evaluations, the authors propose two strategies for updating datasets: a mimicking strategy that generates samples similar to the original data and an extending strategy that broadens existing samples by incorporating Bloom's taxonomy. Through extensive experiments on MMLU and BIG-Bench, the paper demonstrates the effectiveness of these strategies in maintaining benchmark stability and mitigating issues related to benchmark leakage and difficulty control, thereby enhancing the reliability of model evaluations.

---

> ### Author Rebuttal · Authors · 2024-08-22
>
> Response to Reviewer miBT
>
> Thanks for your review and suggestion. It appears there may have been some misunderstandings regarding the contribution, and the experiment of our work, and some inquiries were not clearly articulated. Based on our interpretation, we have tried to address these questions as comprehensively as possible. We are open to and look forward to further discussion to clarify any concerns.
>
> - **Regarding the concerns "datasets used are relatively small" and "generalizability of the proposed method may be questionable"**
>
> **Response 2.1:** We believe the framework is generalizable to larger-scale datasets. For "Math having only 1000 instances" is due to time and cost considerations. We use the first 1000 samples from the Math dataset as seed samples and deliberately control the update process to produce only size-matched new samples.
>
> To address your concerns, we mimic the whole Math dataset and generate 6000 samples. We test the Llama-7b model, which performs with an original dataset score of 15.8 and 16.0 on the mimicked dataset. This demonstrates the method's generalizability to a larger-scale dataset.
>
> Considering that our auto-update process is based on existing benchmarks, and recognizing that most popular evaluation benchmarks typically feature subtask datasets of around 1k instances or even less (e.g., GSK8k: 1319, MBPP: 499, ARC Challenge: 1172). However, we are open to expanding our experiment on the larger-scale dataset to demonstrate generalizability.
>
> - **Regarding "does not discuss the distinctions and connections with existing data leakage work".**
>
> **Response 2.2:**  Most existing research focuses on detecting data leakage. To the best of our knowledge, our work is the first work to auto-update the dataset to alleviate overestimation caused by data leakage.  We appreciate your suggestion and will dedicate an additional page in our paper to discuss how our approach differs from and connects with existing methodologies.
>
> - **Regarding the suggestion "Could a specific example be provided to aid understanding?"**
>
> **Response 2.3:**  We have included a detailed example in Figure 1 of the paper. Additionally, further demonstrations are available through the web link and in Appendix E. To further address your concerns, here’s a simplified explanation with examples across different cognitive levels:
>
> --Knowledge – Recall of information: What is the mathematical formula for the Law of Universal Gravitation, who formulated it?
>
> --Comprehension – Understanding concepts: Explain the main idea of Newton's first law.
>
> --Application – Applying knowledge in different contexts: If two celestial bodies, one with a mass of 5x10^24 kg and the other with a mass of 6x10^24 kg, are initially 1x10^6 km apart, how would the gravitational force between them change if the distance between them is reduced by half?
>
> --Analysis – Breaking down information:  Analyze the Law of Universal Gravitation and Coulomb's Law. How do these laws compare and contrast in terms of their mathematical forms, the forces they describe, and the entities they apply to?
>
> --Create – Creating new ideas or solutions: Write a logically organized essay in favor of euthanasia.
>
> --Evaluation – Judging and critiquing based on established criteria: Question: According to Newton's Law of Universal Gravitation, how does the gravitational force between two objects change if the distance between them is tripled, and how does this principle apply to the interaction between the Earth and the Moon? Answer: According to Newton's Law of Universal Gravitation, if the distance between two objects is tripled, the gravitational force between them is also tripled. This principle applies to the interaction between the Earth and the Moon in such a way that if the Moon were to move three times further away from the Earth, the gravitational pull of the Earth on the Moon would be three times stronger, thus keeping the Moon in its orbit. Is the answer right or wrong?
>
> - **Regarding "The term leakage simulation mentioned in line 198 is unclear"**
>
> **Response 2.4:** Here "leakage simulation" we refer to follows previous research [1] to train the model using the test data. Specifically, we use the test data samples along with the corresponding prompts during the evaluation to construct the training data. This constructed data is then used to train the three selected base models. Thanks for your suggestions, we will make it clearer in the new edition.
>
> - **Regarding Figures 2 and 3,  your concerns "exhibit significant deviations even in the original data" and following questions**
>
> **Response 2.5:** To investigate the consistency of our evaluation results when our mimicking strategy is applied multiple times to the same dataset, we repeated the update process four times. Figures 2 and 3 illustrate the performance on the original dataset (represented by solid bars), the average performance across the four mimicked datasets(represented by dashed bars), and the variation (box plots on the **dashed bars**) across the four mimicked datasets for the selected model.
>
> From "+rationales" and "+leakage" compared to "none" has no significant improvement (e.g. Mistral-7b improved from 59.3 to 61.8 on Medicine), we conclude that: 1:"The mimicking strategy does not introduce significant differences", which validates the effectiveness of our mimicking strategy --- use LLMs to generate samples that closely resemble existing data, thereby preserving the stylistic, contextual essence and difficulty level as outlined in line 39.
>
> 2: Furthermore, it is noted that models without benchmark leakage issues may maintain the performance across the involved original and mimicked datasets. Conversely, models with such issues may have significant performance gaps (Table 3,4). This in turn indicates the reliability of our framework ---- alleviating overestimation when benchmark leakage occurs.
>
> [1] Don’t Make Your LLM an Evaluation Benchmark Cheater

---

> ### Author Rebuttal · Authors · 2024-08-22
>
> - **Regarding your concerns "some tasks show performance degradation, while others show improvement" and "repeating the experiments in Table 4 might show opposite results."**
>
> **Response 2.6:** The observed performance increasing and decreasing are following specific patterns that support our conclusions:
>
> On the Original data:
>
> 1) Pattern 1:  Regardless of the use of Lora, performance improves with "+leakage" compared to "none" (indicated by red cells). e.g. Llama-7b improved from 2.0 to 42.2 on Algos.
>
> 2) Pattern 2:  Performance also improves with "+rationale" compared to "none" (indicated red cells) e.g.  Mistral-7b improved from 27.1 to 54.8 on Element.
>
> 3) Pattern 3: However, performance decreases when compared directly between '+rationale' and '+leakage'. e.g. Llama2-13b have 49.3 (+rationale) but 65.7 (+leakage) on Element task.
>
>  From Pattern 1 we conclude: 1: "Leakage" due to overfitting indeed significantly boosts performance on seen samples. From Pattern 2 & Pattern 3, we have 2: Rationales enhance generalizability, which moderates overfitting, thus slightly reducing performance.
>
>
> On the Mimicked data:
>
> 4) Pattern 4: "+leakage" (using the original testing data for training) does not significantly improve performance over "none" (less than 5\% increase, shown in blue cells), and may even decrease (dark blue cells). e.g. Llama3-8b improved from 59.4 to 61.3 on Econ but decreased from 81.0 to 80 an task Security.
>
> 5) Pattern 5: "+rationales" compared to "none" has no significant improvement (blue bars). e.g. Mistral-7b improved from 59.3 to 61.8 on Medicine.
>
> 6) Pattern 6: "+rationales" compared to "+leakage" improves performance. e.g. Llama2-7b have 72.7 (+rationales) but 70.8 (+leakage) on the Law task。
>
> From Pattern 4 we conclude that 1: "leakage" due to overfitting shows low performance on unseen data and From  Pattern 5 &  Pattern 6 we have 2: Rational can mitigate overfitting issues however have no significant advantage on the mimicked dataset.
>
> These results demonstrate that our mimicking strategy can effectively alleviate overestimation when benchmark leakage occurs --- when severe overestimation occurs using the original dataset, the mimicked dataset can reflect the capabilities of the model.
>
> - **Regarding the reason for no results for Llama3-8b and Mistral-7b with full parameters and "+leakage" settings**
>
> **Response 2.7:** The absence of initial results for Llama3-8b and Mistral-7b under the full parameters and "+leakage" settings is not due to "unsatisfactory results." Rather, it is because of constraints related to time and cost, we only conduct experiments that are necessary. Subsequently, to address your concerns, we have conducted training under these two settings, and the updated results are given below. These findings are consistent with the conclusions in Response 2.6. For example, on the original data: performance improves with "+leakage" compared to "none"  (for Llama3-8b from 76.9 to 93.4 on Law) and on the mimicked dataset, "leakage" does not significantly improve performance over "none" (for Mistral-7b from 26.8 to 26.8 on Algebra), and may even decrease (for Llama3-8b from 81.0 to 73.4 on Security).
>
> These confirm that our mimicking strategy can effectively alleviate overestimation when benchmark leakage occurs.
>
> | Model      | Training  |       LoRA       | Algebra $_o$ | Algebra $_m$ | Law $_o$ | Law $_m$ | Econ $_o$ | Econ $_m$ | Medicine $_o$ | Medicine $_m$ | Security $_o$ | Security $_m$ |
> | :--------- | :-------: | :--------------: | :----------: | :----------: | :------: | :------: | :-------: | :-------: | :-----------: | :-----------: | :-----------: | :-----------: |
> | Llama3-8b  |   None    |        -         |     34.0     |     36.5     |   76.9   |   82.0   |   50.9    |   59.4    |     65.9      |     70.3      |     74.0      |     81.0      |
> | Llama3-8b  | + leakage | $\boldsymbol{x}$ |     50.0     |     22.6     |   93.4   |   74.3   |   64.9    |   55.6    |     78.1      |     67.0      |     84.0      |     73.4      |
> | Mistral-7b |   None    |        -         |     34.0     |     26.8     |   70.2   |   77.1   |   36.9    |   45.7    |     59.5      |     59.3      |     70.0      |     74.0      |
> | Mistral-7b | + leakage | $\boldsymbol{x}$ |     64.0     |     26.8     |   98.4   |   76.5   |   84.2    |   44.6    |     96.0      |     55.5      |     95.0      |     70.0      |
>
>
> - **Regarding the concerns "paper's premise is updating data"**
>
> **Response 2.8:** Thank you for this suggestion. Achieving automating dataset updates is to effectively automate evaluations, and indeed, in some cases where mimicking does not suffice, the extended data may deviate somewhat from the original dataset.
> Moving forward, one possible solution is to further improve the mimicking strategy --- incorporating different entities and additional information to maintain the style of the original dataset while varying the content. We will continue to explore and refine our framework.
>
> Although the "mimic" approach preserves the original style, the "extend" approach may change it,  our vision is to achieve scaling up evaluation through automated updates to benchmarks. In this way we can encompass as many styles, distributions, and focusing many capabilities as possible, thus providing a comprehensive evaluation. (Given that progress in evaluation has been relatively slow compared to training scaling up laws, we believe that scaling evaluation is a necessary and likely path forward.)

---

> ### Author Rebuttal · Authors · 2024-08-22
>
> - **Regarding "Section 3.5 not extend the previous setup for experiments on Algebra"**
>
> **Response 2.9:** We did not expand the Algebra and Algorithms in Section 3.5 not due to "unsatisfactory results," but based on a consideration of whether the dataset difficulty was sufficient to effectively differentiate model performance. In Section 3.5, we specifically selected subjects like Sports and Physics because when testing with models such as GPT-4, ChatGPT, and Claude2 the absolute scores are notably high, indicating that the questions are relatively simple and thus insufficient to effectively differentiate between the models' capabilities. For Algebra and Algorithms, the existing data difficulty is already sufficient to differentiate between the models, so we do not extend it.
>
> - **Regarding why "significant performance improvement with more popular seeds"**
>
> **Response 2.10:** As outlined in line 252, previous research [2] indicates that the popularity of a subject entity correlates inversely with the difficulty of the questions associated with it. Therefore, as the popularity of the seeds increases, the performance becomes higher.
>
> - **Regarding "Why does Table 6 lack results under the "+leakage" setting?"**
>
> **Response 2.11:** As the first work to explore solutions to benchmark leakage, we conduct an extensive range of experiments. Given the limitations of computational resources, we only conduct experiments that are necessary.
> The "+leakage" setting has already been shown to cause severe overfitting problems, as discussed in Sections 3.3 and 3.4. Thus, including it in Table 6 is not imperative for showing our method's effectiveness.
>
>
> However, to address your concerns, we conduct experiments under the "+leakage" settings for mistral on the extended Algebra dataset. Given the limited time available for rebuttal, we are open to including more results in the subsequent updates if necessary. The results show a decline in model performance on our extended dataset. This shows the adaptability of our framework to utilize different language models as backbones.
>
> | Model      |   Training    |       LoRA       | Algebra (Claude) |
> | :--------- | :-----------: | :--------------: | :--------------: |
> | Mistral-7b |     None      |        -         |   **4 5 . 5**    |
> | Mistral-7b |   +leakage    |   $\checkmark$   |       5.4        |
> | Mistral-7b | + w rationale |   $\checkmark$   |       17.9       |
> | Mistral-7b |   +leakage    | $\boldsymbol{X}$ |       4.5        |
> | Mistral-7b | + w rationale | $\boldsymbol{X}$ |       6.8        |
>
> - **Regarding the question "if extended datasets generated by GPT-4 would still be effective if leakage occurs.**
>
> **Response 2.12:**   As mentioned in Response 2.11, due to the constrained timeline for the rebuttal, we adopted a uniform setting to address all questions. We train the Mistral model using extended datasets generated by GPT-4. Subsequently, we use the extended dataset by Claude-3 to conduct evaluation.  We plan to expand upon additional settings and datasets in future updates.
>
> The result shows that the comparison between the scores, 17.3 and 45.5, confirms that training on GPT-4 versions of the test data did not lead to a severe overestimation on test data by Claude3. Additionally, In our previous works, we have proposed a framework that employs multiple large models to mimic peer review processes. Moving forward, our focus indeed is to continue to use it for better evaluation.
>
> | Model      |      Training      |     LoRA     | Algebra (Claude) |
> | :--------- | :----------------: | :----------: | :--------------: |
> | Mistral-7b |        None        |      -       |   **4 5 . 5**    |
> | Mistral-7b | Trained with GPT-4 | $\checkmark$ |       17.3       |
>
>
> - **Regarding the suggestion "the Related Work section is somewhat scattered"**
>
> **Response 2.13:** Please refer to the Response 2.2.
>
> - **Regarding the suggestion "There are several minor grammatical"**
>
> **Response 2.14:**  Thanks for your suggestion,  we may revise it in the new edition.
>
> [2] When not to trust language models: Investigating effectiveness of parametric and non-parametric memories.

---

> ### Author Response · Authors · 2024-08-26
> **Looking forward to your feedback**
>
> Dear reviewer miBT,
>
> Thank you again for your helpful reviews. As the end of the discussion period draws near, we kindly request for your feedback and comments. If you find that our responses have effectively addressed your concerns, can you give us feedback? Your feedback is crucial to us. We apologize for any inconvenience caused.

---

> > ### Comment · Reviewer_miBT · 2024-08-27
> > **Thank you for your response**
> >
> > You solve most of my concerns, I will raise my final rating for this study.
> >
> > Just a quick question—how should the data be updated if I want to apply this method to classification tasks or dialogue generation tasks? It currently seems challenging to extend it to other tasks.

---

> > > ### Author Response · Authors · 2024-08-30
> > > **Response to Reviewer miBT**
> > >
> > > Thank you for raising the score.
> > >
> > > Regarding these two tasks, the mimic strategy easily updates the data. For example, Sports (BBH) is a classification task.

---

### Official Review · Reviewer_Eypk · 2024-08-02
**Important problem**

**Rating:** 6
**Confidence:** 3
**Correctness:** The claims are correct with solid exp…
**Clarity:** The paper is well written with inform…

**Review:**

Strengths:
- A very important topic as most benchmarks are leaked or saturated
- It shows the proposed method can alleviate the overestimating problem

Weakness:
- The answers are generated by LLMs as well. How do you make sure the generated answers are correct?
- This method can possibly be attacked because people can use similar methods to generate synthetic data and improve the benchmark scores.

**Strengths:**

see above.

**Additional Feedback:**

N/A

**Documentation:**

N/A

**Limitations:**

The paper briefly mentioned some limitations but it will be good to see a whole section to discuss the limitations of this method.

**Opportunities For Improvement:**

Instead of fully using LLM to generate the answers, you can try to generate some golden answers with labelers or other tools. You can then  check whether the newly generated answers are mostly correct.

**Relation To Prior Work:**

Yes

**Summary And Contributions:**

This paper introduces a framework for automating the updating of datasets for evaluating large language models (LLMs), addressing the issues of benchmark leakage, difficulty control, and evaluation stability. The proposed strategies include the "mimicking" strategy, which generates new samples similar to the original data, and the "extending" strategy, which expands samples at various cognitive levels using Bloom's taxonomy. Extensive experiments demonstrate the effectiveness of these strategies in mitigating overestimation from benchmark leakage and providing fine-grained performance analysis.

---

> ### Author Rebuttal · Authors · 2024-08-22
>
> Response to Reviewer Eypk
>
> Thank you for your careful review and helpful suggestions. Here are the responses to the questions:
>
> - **Regarding how to "make sure the generated answers are correct"**
>
> **Response 1.1:** To ensure the generated answers' quality, we use a heuristic way to filter generated samples (as mentioned in lines 79-81 and Appendix A). The human evaluation result on the filtered examples shows that the correctness of the answer for the mimicking and extending data is 86.7 and 92.7 respectively, and the agreement is 82.8 and 82.0 respectively (Table 2).
>
> The detailed heuristic way includes:
>
> --Math Word Problems with Hints: we verify the arithmetic expression (for example the calculation formula divide(add(120, 70), 2) for the question: "The speed of a car is 120 km in the first hour and 70 km in the second hour . what is the average speed of the car?") generated with the Q&A pair to ensure it matches the final answer.
>
> --Computer Science Algorithms: we use the periodic table to verify the correctness of the element provided in the answer.
>
> --Programming Tasks: we employ written programs to check if the answer strings are correct.
>
> --Other Tasks: we deploy GPT-4 as the judge [1] to conduct evaluations for the answers.
>
>
> - **Regarding the worry about "method can possibly be attacked"**
>
> **Response 1.2:** Thank you for your suggestions, which align with the core objectives of our research. To our knowledge, this paper is the first work aimed at addressing this issue. Unlike static datasets, our method enables timely and automatic updates of datasets to prevent performance boosts that stem from overfitting to test samples. Our results indicate that the strategies we propose can effectively mitigate such negative impacts by updating the dataset. We will also going to explore potential attacks that could challenge our method, and how to detect and respond to such attacks by promptly updating our dataset in future work.
>
> - **Regarding the suggestion "generate some golden answers with labelers or other tools"**
>
> **Response 1.3:** Thank you for your suggestion. We are currently exploring the use of additional tools to help ensure the accuracy of answers generated in our framework.
>
> [1] Judging LLM-as-a-Judge with MT-Bench and Chatbot Arena

---

> ### Author Response · Authors · 2024-08-26
> **Looking forward to your feedback**
>
> Dear reviewer Eypk,
>
> Thank you again for your helpful reviews. As the end of the discussion period draws near, we kindly request for your feedback and comments. If you find that our responses have effectively addressed your concerns, can you give us feedback? Your feedback is crucial to us. We apologize for any inconvenience caused.

---

### Decision · Program_Chairs · 2024-09-26

**Decision:**

Accept (Poster)

**Comment:**

This paper addresses the challenge of detecting structural similarity between queries for in-context learning models. The authors introduce a model designed to estimate structural similarity accurately, trained on a high-quality dataset of 170k question pairs. The model's effectiveness is validated through significant improvements in Kendall-Tau distance and precision@k metrics, underscoring the model's potential to enhance in-context learning through better structural similarity detection.

Strengths:
1. This paper is well-organized and easy to read.
2. The research topic is interesting, and the proposed method is instructive.
3. The experimental results are comprehensive, and the performance met expectations initially.

Weaknesses:
1. This method may be vulnerable to exploitation, as individuals could employ similar techniques to generate synthetic data and artificially enhance benchmark scores.
2. It would be better to compare the proposed methods with previous methods for alleviating data leakage.
3. A more detailed ablation study regarding the four extending strategies should be conducted.